# Assessment of the efficacy and safety of two albendazole regimens for the treatment of hypermicrofilaraemic loiasis in adults in Woleu-Ntem Province, Gabon: A phase IIb single-blind randomised controlled trial

**Noé Patrick M'Bondoukwé**⊙*, **Luccheri Ndong Akomezoghe,**
**Bridy Chesly Moutombi Ditombi, Jacques Mari Ndong Ngomo,**
**Hadry Roger Sibi Matotou, Ginette Severine Zang Ondo, Valentin Migueba,**
**Coella Joyce Mihindou, Bedrich Pongui Ngondza, Christian Mayandza,**
**Héléna Noéline Kono, Dimitri Ardin Moussavou Mabicka, Charleine Manomba,**
**Reinne Moutongo, Luice Aurtin Joel James, Denise Patricia Mawili Mboumba,**
**Marielle Karine Bouyou Akotet**

Department of Parasitology-Mycology-Tropical Medicine, Faculty of Medicine, Université des Sciences de la Santé, Libreville, Gabon

* mbondoukwenoe@gmail.com

## Abstract

### Background

*Loa (L.) loa* hypermicrofilaraemia (≥ 8,000 mf/mL) increases the risk of severe adverse events during mass ivermectin administration for onchocerciasis control. Albendazole has been proposed as a potential alternative for reducing microfilaraemia prior to ivermectin administration.

### Methodology and principal findings

This prospective study was conducted in northern Gabon from November 2021 to April 2022. Individuals infected with *L. loa* were screened and allocated to three groups: two treatment arms receiving 400 mg or 800 mg of albendazole daily for 30 days among hypermicrofilaraemic participants, and a group with microfilaraemia < 8,000 mf/mL. Clinical symptoms and parasitological data were collected on Days 0, 2, 7, 14, and 30. A total of 70 participants were enrolled: 16 in the 400 mg group, 16 in the 800 mg group, and 38 in the group with microfilaraemia below 8,000 mf/mL. Itching was the most frequently reported adverse event. By Day 30, no participants in the group with microfilaraemia below 8,000 mf/mL presented with clinical symptoms. Microfilaraemia significantly decreased in all groups (p < 0.01). After 30 days, over 70.0% of patients treated with albendazole had microfilaraemia < 8,000 mf/mL. There was no significant difference in efficacy between the two albendazole regimens.

**Data availability statement:** All relevant data are in the manuscript.

**Funding:** This project is funded by the European and Developing Countries Clinical Trials Partnership (EDCTP) under the TMA (Training and Mobility Actions) 2019 Career Development Fellowship (CDF) – Grant No. TMA2019CDF-2730, on the treatment of hypermicrofilaraemic loiasis and the evaluation of two albendazole protocols. The fellowship was awarded to N.P.M. The funders had no role in study design, data collection and analysis, decision to publish, or preparation of the manuscript.

**Competing interests:** The authors have declared that no competing interests exist.

## Conclusions/Significance

Daily administration of 400 mg albendazole for 30 days effectively reduces microfilarial loads in patients with *L. loa* hypermicrofilaraemia and is well tolerated and safe. This pre-treatment regimen may reduce the risk of adverse events associated with ivermectin administration. Further research is needed to evaluate the long-term persistence of microfilarial suppression.

## Author summary

Loiasis is a parasitic disease caused by the *Loa loa* worm and transmitted by the bite of a fly called *Chrysops*. People living in Central Africa can carry millions of these microscopic worms in their blood. However, treating this disease is complicated: when the number of parasites in the blood is too high, giving the usual treatments (like ivermectin) can lead to dangerous side effects, including coma or even death. This study tested whether albendazole, a common anti-parasitic drug, could safely reduce the number of worms in heavily infected patients in rural Gabon. We compared two doses - 400 mg and 800 mg taken daily for 30 days - and found that both doses were effective in lowering parasite levels below the danger threshold. The lower dose worked just as well and was better tolerated, making it more suitable for use in individual-scale treatment campaigns. Our findings suggest that a 400 mg daily regimen of albendazole may help prepare patients for safer treatment with curative medications and could improve control of this neglected disease in hard-to-reach communities.

## Introduction

In areas where onchocerciasis is co-endemic with *Loa loa*, the mass drug administration (MDA) strategy using ivermectin (IVM) for onchocerciasis control is hampered by the presence of hypermicrofilaraemic individuals carrying more than 8,000 microfilariae (mf) per mL of blood [1]. These individuals are at higher risk of developing severe adverse reactions (SARs) following IVM administration. Moreover, the severity of these adverse effects increases with parasitic load: individuals with microfilaraemia exceeding 30,000 mf/mL are at up to 200-fold higher risk, and those with more than 50,000 mf/mL at up to 1,000-fold higher risk of developing serious or fatal outcomes, compared with individuals with < 8,000 mf/mL [1,2]. The probability of SARs after IVM administration were recently evaluated in Cameroon varying greatly according to the number of microfilariae found in the blood: the risk is 0.1% for individuals with 8,000 mf/mL and around 37.0% in those with 100,000 mf/mL [3]. The prevalence of *L. loa* microfilaraemia exceeds 40.0% in some villages in onchocerciasis transmission areas (PNLMP: Programme National de Lutte contre les Maladies Parasitaires).

Currently, no safe and effective treatment is recommended for hypermicrofilaraemic loiasis. Consequently, patients with such high parasitic densities are often left

untreated when alternative non-pharmacological strategies, such as vector control measures (e.g., insecticide spraying, larval habitat management) or physical protection with nets, are not available. In the "test and treat" strategy, the entire population is screened to identify hypermicrofilaraemic individuals; those with high microfilaraemia are excluded from large-scale IVM administration, thereby maintaining the reservoir of *L. loa* within the community [4].

A treatment allowing a reduction of hypermicrofilaraemia below the 8,000 mf/mL threshold for a sufficiently long period would enable subsequent IVM administration and may contribute to the control and elimination of both onchocerciasis and lymphatic filariasis (LF) in areas co-endemic with *L. loa*. The World Health Organization (WHO) has suggested semiannual treatment with albendazole (ALB), complemented by vector control measures, as a potential strategy for LF elimination in *L. loa* transmission areas [5]. This molecule could also be used to reduce hypermicrofilaraemia in individuals living in well-defined onchocerciasis-endemic foci prior to mass IVM administration (IVM MDA). For more than 30 years, albendazole has been considered an alternative to diethylcarbamazine (DEC) or IVM for the curative treatment of *L. loa* [6]. Previous placebo-controlled studies, such as that of Klion et al. reported that albendazole administered for 21 days (400 mg twice daily) led to partial reductions in *L. loa* microfilaraemia in some, but not all, hypermicrofilaraemic patients [7]. These findings suggested that albendazole may progressively reduce microfilarial loads and thus help prepare patients for safer subsequent therapy. It has been reported that administering 2–6 doses of 800 mg albendazole, at two-month intervals, can reduce *L. loa* microfilaraemia by at least 50.0% and potentially maintain parasitaemia below 8,000 mf/mL for at least 4 months [8].

In Gabon, DEC is no longer available. There is a need to provide evidence-based information to support national and international recommendations for managing hypermicrofilaraemic *L. loa* using approaches that exclude both IVM and DEC. Moreover, it is important to use effective molecules for parasitic diseases in the same geographical areas, as recommended by the NTD Roadmap [9]. The Department of Parasitology-Mycology-Tropical Medicine (DPMTM) of the Libreville Faculty of Medicine serves as the reference centre for medical parasitology and maintains one of the largest cohorts of *L. loa* patients. Weekly consultations, along with clinical management for filariasis patients, are conducted at this centre [10]. Albendazole has been routinely used for the curative treatment of *L. loa* since the interruption of DEC and the Mectizan programme over the past 10 years, based on routine clinical practice data. Recently, DPMTM reported that administration of 400 mg albendazole showed a reduction rate of more than 80.0% in microfilaraemia in a clinical study conducted in Gabon among patients with low microfilaraemia [11]. It is also necessary to demonstrate whether this effective protocol can be applied to hypermicrofilaraemic participants to enable their eligibility for IVM administration.

Thus, the present study aims to identify an optimal treatment regimen for reducing *L. loa* hypermicrofilaraemia in patients living in hyperendemic areas of northern Gabon.

## Materials and methods

### Ethics statement

This study was conducted as part of the PHYLECOG project, financed by the EDCTP2 programme under reference TMA2019CDF-2730. The study protocol was reviewed and approved by the National Ethics Committee for Scientific Research (CNER) (Protocol No. 0053/2022/CNER/P/SG). Written informed consent was obtained from each participant after explaining the study to local and health authorities. The study was registered under the reference ISRCTN14889921.

### Study area

This study was conducted in Gabon, a Central African country bordered to the north-west by Equatorial Guinea, to the north by Cameroon, to the east and south by the Republic of the Congo, and to the west by the Atlantic Ocean (Fig 1). In 2021, its population was estimated at 2.1 million inhabitants. Located in the heart of Africa and straddling the Equator, Gabon is administratively divided into nine provinces, each with distinct geographical and cultural characteristics.

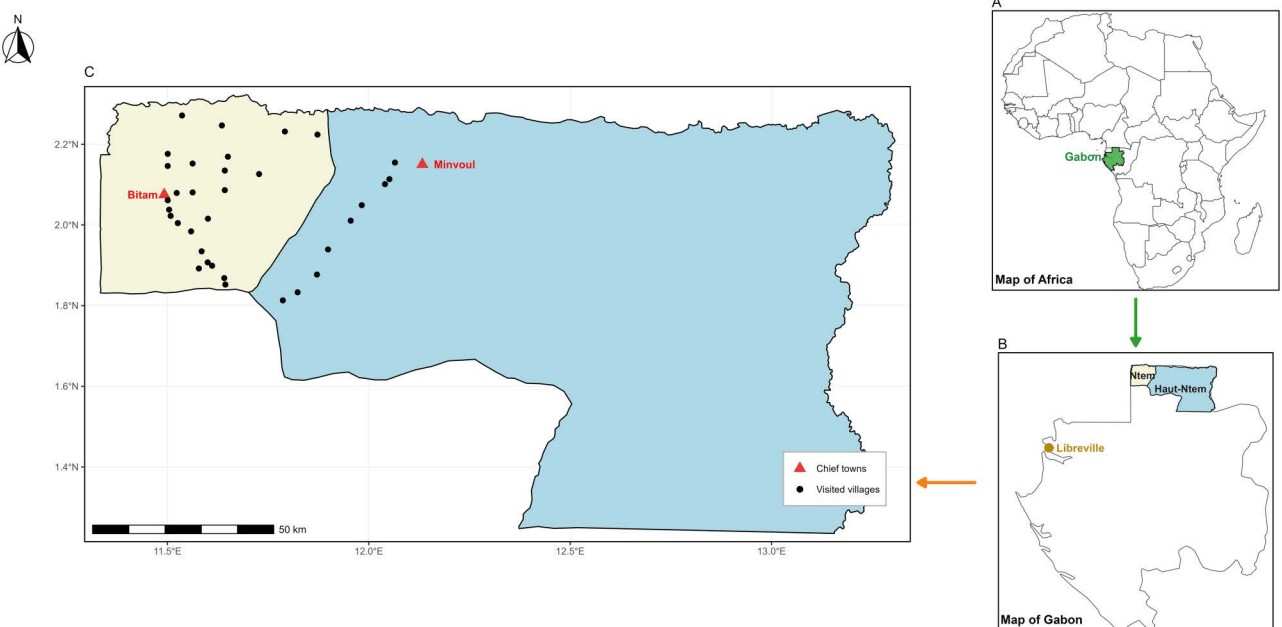

**Fig 1. Study areas. A**: Map of Africa with Gabon highlighted in green. **B**: Map of Gabon with the Ntem (beige) and Haut-Ntem (blue) departments. The yellow marker indicates the location of Libreville, the political capital of Gabon. **C**: Black dots represent the villages where participants were screened and/or recruited. The beige area corresponds to the Ntem department, and the blue area corresponds to the Haut-Ntem department. The red triangles correspond to the capitals of the Ntem and Haut-Ntem departments. Base layers: Natural Earth (public domain;https://www.naturalearthdata.com/about/terms-of-use/). Administrative boundaries (province and department levels): SALB / UN OCHA, accessed via HDX: https://data.humdata.org/dataset/d8d992f3-5374-448f-a3d5-bed7c1925a11/resource/46463ad7-36e0-401d-9608-ef4d360c32fe/download/gabadmbndp21msalb.shp.zip. Village and chief town coordinates: field GPS data collected by the authors (CC BY 4.0).

The country is traversed by several major rivers, most notably the Ogooué River, Gabon's principal watercourse, which spans approximately 1,200 km, draining nearly 80% of the national territory. Among its major tributaries are the Ivindo River, which extends for 500 km, and the Ngounié River, which runs for 300 km.

Gabon stretches about 800 km in length and varies from 20 to 300 km in width, with approximately 80% of its area covered by dense, tropical rainforest. This vast forested region extends from west to east, beginning with a coastal basin characterised by forest-savannah mosaics, progressing into the interior's plateau forests in the north-east, and including a broad mountainous and forested belt of 60–100 km running parallel to the coast. Isolated savannah and shrubland areas are found in the south and south-east of Gabon, contributing to the country's rich and diverse ecosystems [12].

## Study site

This study was conducted in the rural population of Woleu-Ntem, specifically in the departments of Ntem and Haut-Ntem. The region has a population of approximately 155,000 inhabitants, with a population density of about 4 inhabitants per km$^2$. Woleu-Ntem is a province located in northern Gabon. It covers an area of 38,465 km$^2$, and its capital city is Oyem.

The province has a hot and humid equatorial climate with heavy rainfall. It is predominantly covered by secondary forest and boasts a rich and diverse fauna. The prevalence of *L. loa* in the region is estimated at 20.2% [12]. Woleu-Ntem is a relatively under-industrialised area known for producing cocoa, coffee, and rubber. Subsistence farming and hunting in the forest still constitute significant parts of the local diet. The economy of the province revolves around agricultural

produce. There are strong ties with Equatorial Guinea and southern Cameroon, partly driven by shared Fang ethnic heritage among the majority of the population on both sides of the border.

Villages around Bitam (Ntem Department) and Minvoul (Haut-Ntem Department) were selected for this clinical study, where all participants lead a rural lifestyle that exposes them to the *Chrysops* sp. vector of *L. loa* (Fig 1).

**Type and study period.** This was a Phase IIb randomised, single-blind clinical trial conducted from November 2021 to April 2022.

## Study population

Inclusion/exclusion/withdrawal criteria were presented in Table 1, below:

## Sample size calculation

The pwr.anova.test function in R version 4.4.4 was used to determine the sample size for clinical trials where the primary endpoint is a continuous variable that does not necessarily follow a normal distribution. Additionally, the number of treatment arms was set at three. *L. loa* microfilaraemia was the primary endpoint for both hypermicrofilaraemic participants and the group with microfilaraemia below 8,000 mf/mL.

The following formula, based on Cohen's guidelines, was applied, where "k" denotes the number of comparison groups and "f" the effect size, with Cohen's f values of 0.1, 0.25, and ≥ 0.4 representing small, medium, and large effect sizes, respectively.

Considering three treatment arms (k = 3), a large effect size (f = 0.6), an alpha risk of 5%, and a statistical power of 80%, a minimum of 10 patients completing treatment in each group would be required to ensure sufficient power to validate the results.

## Study design

The study design is illustrated in the Fig 2 below.

**Study drugs.** The investigational product was Ubigen albendazole 400 mg (UBITHERA PHARMA PVT. Ltd.). The tablets were administered orally, 15–30 minutes after a fatty meal (such as fatty rolls with approximately 15 g of butter), to optimise absorption and efficacy. All participants also received a 10 mg cetirizine tablet daily for seven days. Participants with microfilaraemia below 8,000 mf/mL received a single daily dose of 400 mg albendazole for 30 days, whereas those

**Table 1. Inclusion, exclusion, and withdrawal criteria for trial participants.**

| Criteria | | |
|---|---|---|
| **Inclusion** | **Exclusion** | **Withdrawal** |
| Age between 18 and 75 years; | Having declared chronic diseases such as HIV/AIDS, acute or chronic hepatitis, cancer, diabetes, chronic heart disease, or renal disease; | Withdrawal of consent to participate in the study; |
| Weight less than 90.0 kg; | Taking anthelmintics such as diethylcarbamazine (DEC), suramin, ivermectin, mebendazole, or albendazole within four (4) weeks prior to screening; | Worsening of symptoms or the appearance of severity criteria related to *L. loa* or another disease; |
| *L. loa* microfilaraemia (≥8,000 mf/mL for the treatment group and <8,000 mf/mL for the comparison group); | Being allergic to benzimidazoles; | An increase in microfilaraemia of more than 50% compared to the initial value. |
| Signed an informed consent form; | Being pregnant or breastfeeding | |
| Agreement to comply with the study procedures, including blood sample collection; | | |
| Residing in or being in the village during the clinical trial period. | | |

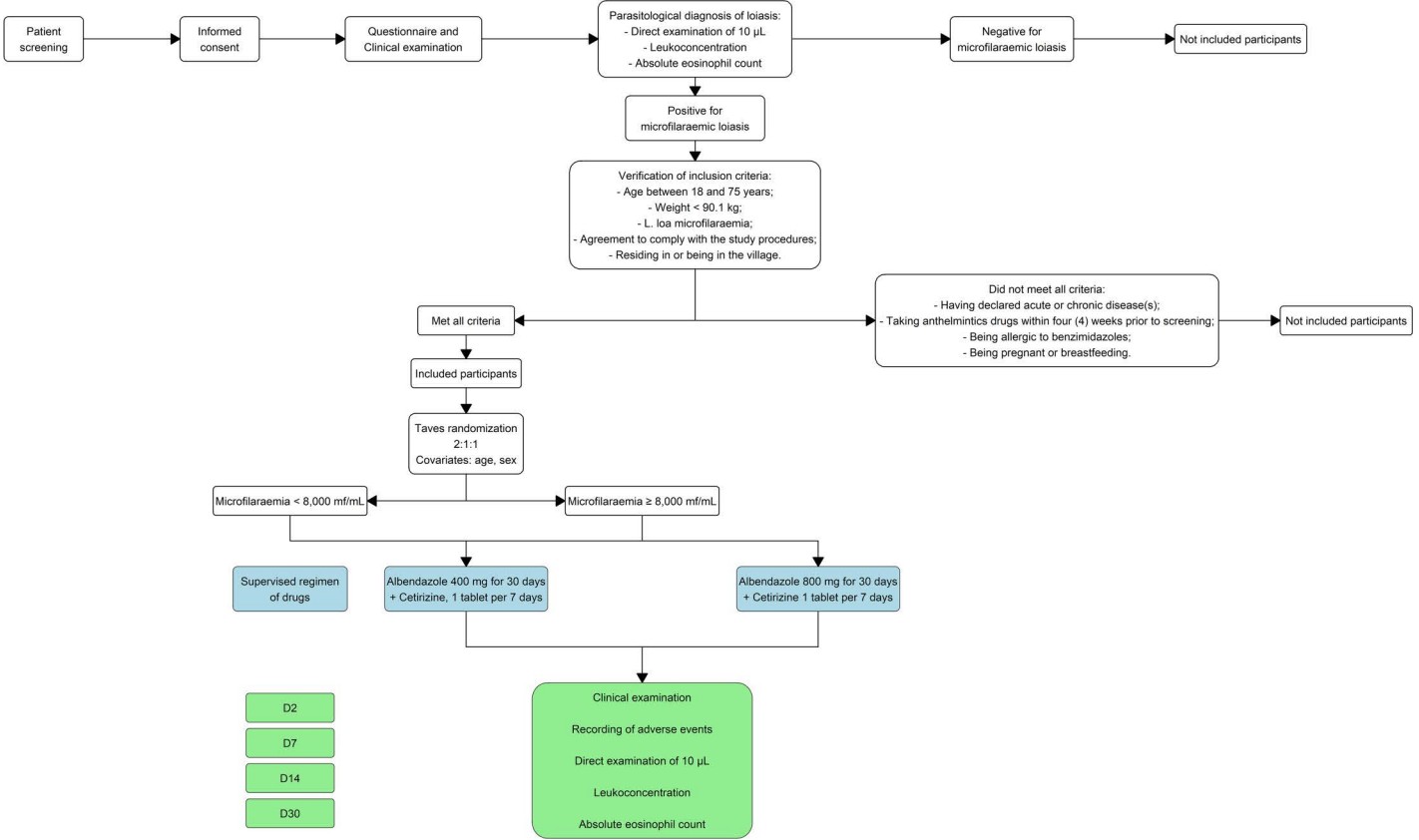

**Fig 2. Study design.** This figure corresponds to different steps ranging from the screening of participants for the clinical trial to different drug administration (in blue) and procedures at each time point (in green).

with microfilaraemia ≥ 8,000 mf/mL received either 400 mg or 800 mg (two 400 mg tablets) once daily for 30 days. All treatments were administered under the supervision of a designated community worker within the village to ensure proper follow-up. The field team consisted of parasitologists, lab technicians, and field workers who conducted a three-day training session on standardised clinical examination, symptom description, and sample collection techniques for the other study team members, in order to ensure consistency in data collection.

**Safety parameters.** Safety variables included clinical signs and symptoms related to *L. loa* infection, both common and uncommon, such as myalgia/arthralgia, headaches, asthenia, pruritus, transient nerve paralysis, subcutaneous or subconjunctival migration of adult worms, vision disorders, and Calabar swelling. The frequency of adverse events (AEs) up to one month after the first dose of albendazole was monitored and classified according to type, severity, and relationship to the investigational drug, as assessed by the investigators.

Severity was classified as:

- *Mild*: The event is tolerated by the participant, causing minimal discomfort and not interfering with daily activities;

- *Moderate*: The event is uncomfortable enough to interfere with daily activities;

- *Severe*: The event prevents normal daily activities;

- *Not applicable*: Events for which the intensity cannot be meaningfully assessed.

PLOS Neglected Tropical Diseases

Serious adverse events (SAEs) were evaluated according to the ICH definition: any untoward medical occurrence that, at any dose, results in death, is life-threatening, requires hospitalisation or prolongs an existing hospitalisation, results in persistent or significant disability/incapacity, or is a congenital anomaly/birth defect, and is related to any dose of a drug or the doses normally used in humans [1].

**Data collection and screening.** A preliminary visit was conducted to inform local authorities about the study's objectives. After obtaining informed consent from participants, sociodemographic and clinical data were collected, including age, gender, weight, height, and clinical manifestations attributable to *L. loa*. Subsequently, a 4 mL venous blood sample was collected in an EDTA tube, between 10:00 am and 2:00 pm, for parasitological screening of *L. loa* via direct examination of 10 µL of blood and the leukoconcentration technique.

**Patient randomisation.** Taves' covariate-adaptive randomisation was used to allocate participants across the different treatment arms [13]. The covariates included gender (male, female), age groups (18–64 years, 65 years and older), and microfilaraemia strata (less than 8,000 mf/mL, 8,000–30,000 mf/mL, 30,001–50,000 mf/mL, and over 50,000 mf/mL). Patients were randomised according to a 2:1:1 ratio. The list of eligible participants by stratum was provided to an independent pharmacist, who was not involved in the study. The pharmacist was responsible for assigning the treatment indicated on the randomisation list based on the participant's position on the eligible list for each stratum. Additionally, the pharmacist administered the first dose of treatment.

**Patient follow-up.** After the screening, individuals were re-examined after an interval of 12–24 hours. Following the administration of the first dose, the subsequent treatment was supervised by the study team focal points, together with community health workers selected from within the villages. These community workers had been previously trained on all the study procedures. The treatment was taken 15–30 minutes after a fatty meal (such as fatty rolls with approximately 15 g of butter). On scheduled visit days and in villages without a community worker, a mobile team comprising the study physician and/or a pre-trained field health worker responsible for treatment administration, identification, and reporting of *L. loa*-related symptoms, as well as detection, management, and notification of potential adverse events of albendazole, supervised the treatment. During these visits, venous blood samples in EDTA tubes were also collected to monitor microfilaraemia at D2, D7, D14 and D30. No reassessment of the microfilaraemia was performed after Day 30. The participants received ivermectin when they were positive at D30. Efforts were made to examine the participants at the same time (hour, minute) of the day during each follow-up visit, in order to minimize the impact of the diurnal periodicity of microfilaraemia.

### Parasitological diagnosis of *Loa loa* microfilaraemia

Direct blood examination and leukoconcentration techniques allowed the determination of microfilaraemia. Leukoconcentration was used in addition to direct examination of 10 µL of blood to detect *L. loa* microfilaraemia below the detection threshold of the direct examination, i.e., densities lower than 100 mf/mL.

**Direct examination of 10 µL of blood.** A direct examination was performed to detect and quantify microfilariae, thereby assessing the presence and intensity of infection. The slide examination was performed immediately after sample collection to maximize parasite motility, which is essential for species identification. The entire 10 µL of blood were carefully placed on a microscope slide and examined under a microscope (magnification ×10). Parasitaemia was expressed as the number of microfilariae per millilitre of blood (mf/mL), providing quantitative data for infection assessment.

**Determination of microfilaraemia in individuals with high parasite densities.** On average, 30 microscopic fields are examined to cover the entire 10 microlitres of fresh blood during the microscopic direct examination. When more than 10 microfilariae per field were observed (corresponding to approximately 30,000 microfilariae per millilitre), a tenfold (1/10) dilution was performed. In cases of high parasitaemia, a series of tenfold dilutions was carried out until approximately 10 microfilariae per field were observed. The parasitaemia was then calculated according to the number of tenfold dilutions,

using the following formula: $10^n \times (100 \times$ number of microfilariae), where $n$ is the number of tenfold (1/10) dilutions. For high microfilaraemia, two read are performed for minimise the variability in discordant results.

Microscopic examination of each slide, including those prepared after serial dilutions, was performed according to a standardised protocol using a fixed blood volume of 10 μL, systematic field examination, and predefined dilution thresholds. Slides with high microfilaraemia requiring dilution were independently examined by two experienced microscopists, who were blinded to each other's results. Microfilarial density was expressed as microfilariae per 10 μL of blood (mf/10 μL), and the mean value of the two readings was retained for analysis. When a discrepancy greater than 5 mf/10 μL or a borderline count was observed, the slide was reviewed by a third independent microscopist; in such cases, the final microfilarial density corresponded to the mean of the two closest values.

**Leukoconcentration of 4 mL of blood.** The concentration technique was performed according to the method described by Ho Thi Sang and Petithory (1964), which offers increased sensitivity for detecting microfilariae [14]. A volume of 4 mL of peripheral blood was centrifuged at 2,000 rpm for 5 minutes. After centrifugation, the plasma was carefully removed using a pipette. Then, 2 mL of physiological saline (0.9% NaCl) and 1 mL of a 2% saponin solution were added to the red cell pellet, and the mixture was homogenised to ensure complete lysis of red blood cells. A 10-minute waiting period was observed to allow haemolysis. Following this, the tube was centrifuged again at 2,000 rpm for 10 minutes, and the supernatant was discarded. The NaCl and saponin procedure was repeated before the final centrifugation. The resulting white leukocyte pellet was washed with 3 mL of 0.9% NaCl. Once free of debris, the leukocyte pellet was placed between a slide and a coverslip and examined under an optical microscope using a × 10 objective for locating microfilariae and a × 40 objective for identification and quantification.

## Quantification of eosinophilia

Eosinophilia rate was determined from the thin blood smear of 5 μL of blood by calculating the percentage of eosinophils among 100 leukocytes counted.

## Statistical analyses

Data processing was carried out using REDCap, and all statistical analyses were performed with R software version 4.4.2. A significance level of 5% was applied to all tests.

Categorical variables, including sex, symptoms, and adverse events, were presented as frequencies and percentages, and group comparisons were made using the Chi-square or Fisher's exact test, as appropriate. Quantitative variables were summarised using the geometric mean (95% confidence interval, CI) and the median (25th-75th percentiles) to account for data skewness.

Microfilaraemia was expressed as both the geometric mean (95% CI) and median (interquartile range), while eosinophil counts and other continuous parameters followed the same descriptive approach. The microfilaraemia reduction rate was calculated as a percentage using the following formula:

$$\text{Reduction Rate} = \frac{\text{Microfilaraemia at Day 0} - \text{Microfilaraemia during follow} - \text{up}}{\text{Microfilaraemia at Day 0}} \times 100$$

Reduction rates were expressed as median values with their 95% confidence intervals and compared across visits (D2, D7, D14, and D30) and treatment groups using the non-parametric Mann-Whitney U test. Proportions of participants achieving the primary endpoint, microfilaraemia < 8,000 mf/mL, were compared across visits using the Chi-square test. Both intention-to-treat (ITT) and per-protocol (PP) analyses were performed to ensure robustness of the findings.

For each treatment arm, temporal variations in microfilaraemia were evaluated using paired comparisons between baseline (D0) and subsequent visits. Evolution curves were generated to depict both geometric mean trends (with 95% CI) and median values (with interquartile ranges).

Survival analyses based on Kaplan-Meier estimates were performed to describe the probability of maintaining microfilaraemia ≥ 8,000 mf/mL over the 30-day follow-up period. Comparisons between the 400 mg and 800 mg albendazole regimens were conducted using the log-rank test.

All graphs, including microfilaraemia evolution, reduction rates, and survival curves, were produced using the *ggplot2* and *survival* packages in R.

## Results

### Patients

Among the 1,342 volunteers screened for *L. loa* microfilaraemia, 70 participants met all the inclusion criteria for the clinical trial. Of these, 38 had microfilaraemia below 8,000 mf/mL, and 32 had microfilaraemia above 8,000 mf/mL. Participants in the group with microfilaraemia below 8,000 mf/mL received 400 mg of albendazole (ALB) daily for 30 days; 16 participants with microfilaraemia at or above 8,000 mf/mL received 400 mg ALB daily for 30 days, and the remaining 16 received 800 mg ALB daily for 30 days (Fig 3). The number of individuals assessed varied across time points (D0, D2, D7, D14, and D30). There was only one withdrawal of consent in the group with microfilaraemia below 8,000 mf/mL and none in the treatment arms.

Table 2 summarises the characteristics of the study population prior to treatment. There were no significant differences between the three treatment groups for variables such as sex, age, weight, height, and eosinophil count. Regarding the mean microfilarial load, it was comparable between the two treatment arms of hypermicrofilaraemic participants (p = 0.7).

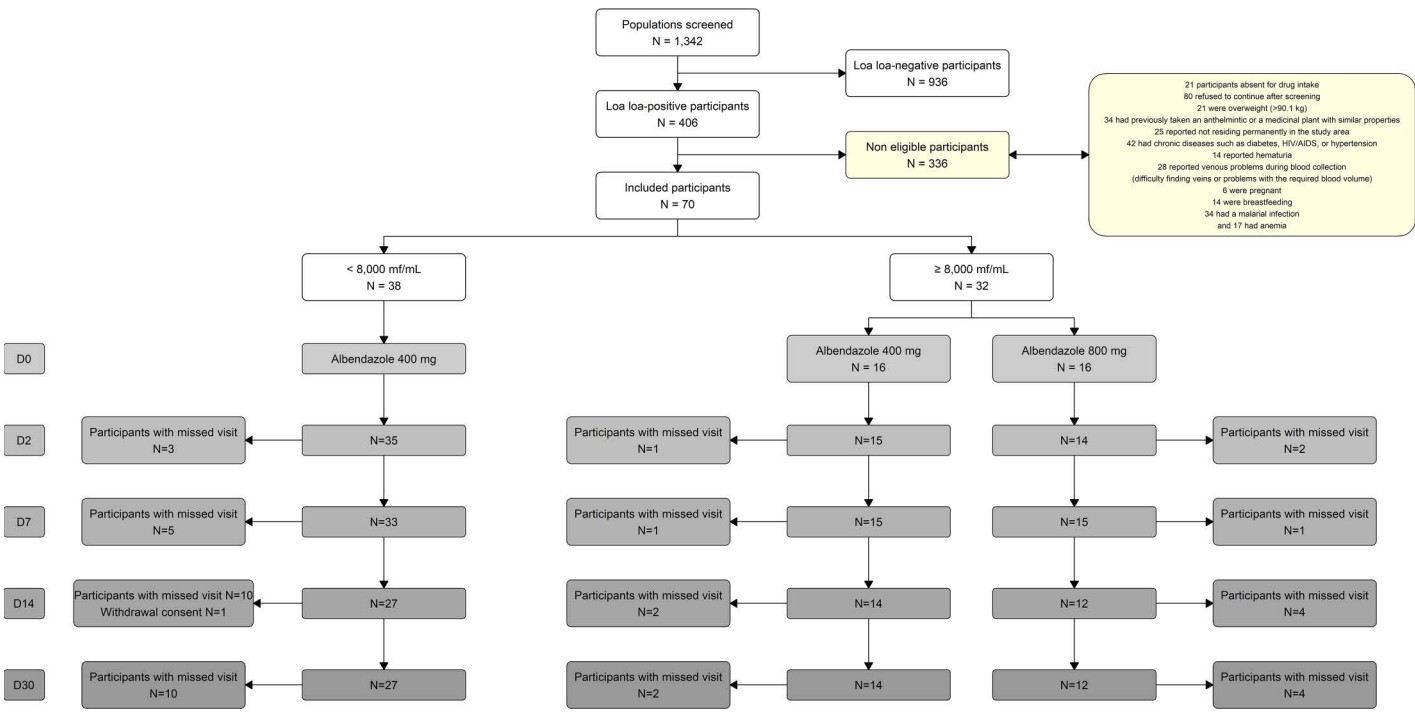

**Fig 3. Study flowchart.** Flow diagram showing the number of participants screened and enrolled in the clinical study. One participant in the group with microfilaraemia below 8,000 mf/mL withdrew consent. Some participants were not present at specific visits ("missed visit"), but they were not considered lost to follow-up, since the study design allowed flexibility with assessments at non-consecutive time points (e.g., inclusion and end-of-treatment visits). The yellow colour corresponds to non-eligible participants and related information for non-inclusion. Each visit was coded in grey from lighter (D0) to darker (D30).

**Table 2. Descriptive characteristics of the treatment groups at the inclusion.**

| Variables | Group with microfilaraemia below 8,000 mf/mL (N = 38) | Albendazole 800 mg (N = 16) | Albendazole 400 mg (N = 16) | p-value |
|---|---|---|---|---|
| **Male, n (%)** | 14 (36.8) | 4 (25.0) | 5 (31.2) | 0.7 |
| **Age in years,** mean ± SD[1] | 49.8 ± 12.4 | 52.2 ± 12.4 | 46.7 ± 12.8 | 0.4 |
| **Weight in Kg,** mean ± SD | 64.2 ± 10.4 | 60.2 ± 10.6 | 63.0 ± 10.1 | 0.8 |
| **Height in cm,** mean ± SD | 164.3 ± 8.4 | 167.6 ± 9.3 | 162.4 ± 7.0 | 0.1 |
| **Microfilaraemia in mf/mL,** geometric mean (95% CI[3]) | 2,252 (1,688 - 3,006) | 17,249 (12,139 - 24,511) | 19,178 (13,178 - 27,968) | NA[2] |
| **Eosinophilia rate in %,** geometric mean (95% CI) | 20.4 (3.0 - 27.0) | 15.1 (2.0 - 29.0) | 18.7 (3.0 - 27.0) | 0.7 |

SD[1]: Standard deviation.

NA[2]: Not applicable for the three groups.

95% CI[3]: 95% confidence interval.

**Reduction in microfilaraemia in intention-to-treat and per-protocol analyses.** Compared to the initial parasitaemia (Day 0), both the intention-to-treat (ITT) and per-protocol (PP) analyses demonstrated significant reductions in microfilarial loads across all study groups by Day 30 (p < 0.01). In the group with microfilaraemia below 8,000 mf/mL, the geometric mean (95% confidence interval) decreased from 2,252 (1,688 - 3,006) mf/mL to 258 (80–835) mf/mL in the ITT analysis (−88.5%), and from 1,938 (1,193 - 3,146) mf/mL to 231 (60–901) mf/mL in the PP analysis (−88.1%).

For hypermicrofilaraemic participants treated with 400 mg of albendazole, microfilaraemia decreased from 17,249 (12,139–24,511) mf/mL to 4,177 (2,057–8,483) mf/mL (−75.8%) in the ITT analysis, and from 18,191 (12,268–26,977) mf/mL to 4,177 (2,057–8,483) mf/mL (-77.0%) in the PP analysis. In the 800 mg albendazole group, the geometric mean decreased from 19,178 (13,150–27,968) mf/mL to 1,987 (830–4,757) mf/mL (−89.6%) in ITT, and from 17,991 (10,578–30,599) mf/mL to 2,352 (596–9,289) mf/mL (−86.9%) in PP (Fig 4A and 4B).

Daily reduction trends were consistent between the two analyses. In the group with microfilaraemia below 8,000 mf/mL, reductions (95% confidence interval) in the geometric means on Day 2 were 34.7 (-15.1 - 58.7) % (ITT) and 36.1 (-23.8 - 54.5) % (PP), increasing to 71.1 (46.0 - 79.6) % and 78.3 (56.5 - 84.2) % respectively by Day 7. In the 400 mg group, reductions on Day 2 were 14.0 (-14.1 - 36.0) % (ITT) and 12.5 (-18.9 - 35.6) % (PP), reaching 78.6 (62.0 - 89.1) % (ITT) and 79.7 (62.0 - 89.1) % (PP) by Day 14. The most pronounced reductions were observed in the 800 mg group: 40.9 (-14.9 - 69.2) % (ITT) and 51.5 (-47.0 - 84.0) % (PP) at Day 2, and 89.6 (78.7 - 93.3) % (ITT) and 86.9 (70.3 - 94.3) % (PP) by Day 30 (Fig 4A and 4C).

When considering the medians with interquartile ranges (IQR), similar patterns were observed (Fig 4B and 4D). In the ITT analysis, the median microfilarial density decreased from 2,150 (1,050–5,425) mf/mL at Day 0–400 (200–1,550) mf/mL at Day 30 in the group with microfilaraemia below 8,000 mf/mL. In the PP analysis, the median fell from 1,600 (700–5,950) mf/mL at Day 0–400 (200–1,450) mf/mL at Day 30 in the group with microfilaraemia below 8,000 mf/mL. In the group receiving ALB 400 mg, there was also a decrease in ITT (17,000 (10,675–26,800) mf/mL at D0 to 1,350 (700–3,825) mf/mL at D30) and PP approaches (16,950 (12,800 – 24,975) mf/mL at D0 to 1,950 (675–7,325) mf/mL at D30). However, in the ALB 800 mg treatment arm, in ITT the median microfilaraemia at baseline D0 was 17,000 (10,675–26,800) mf/mL and increased at D2 to 19,050 (7,225–27,000) mf/mL. In the PP analysis, no increase was observed between D0 and D2 and the curve follows the same trend as the other ones (Fig 4B and 4D).

**Comparison of the geometric mean of the reduction rate among hypermicrofilaremic patients according to albendazole regimen 400 mg *versus* 800 mg and to intention-to-treat and per-protocol analyses.** Between-arm comparison showed no statistically significant difference at any time point (D2, D7, D14 or D30), both in ITT and PP analyses (p > 0.05) (Table 3). However, it is important to notice that the geometric mean of reduction rate was always lower in the ALB 400 mg group compared with the ALB 800 mg group.

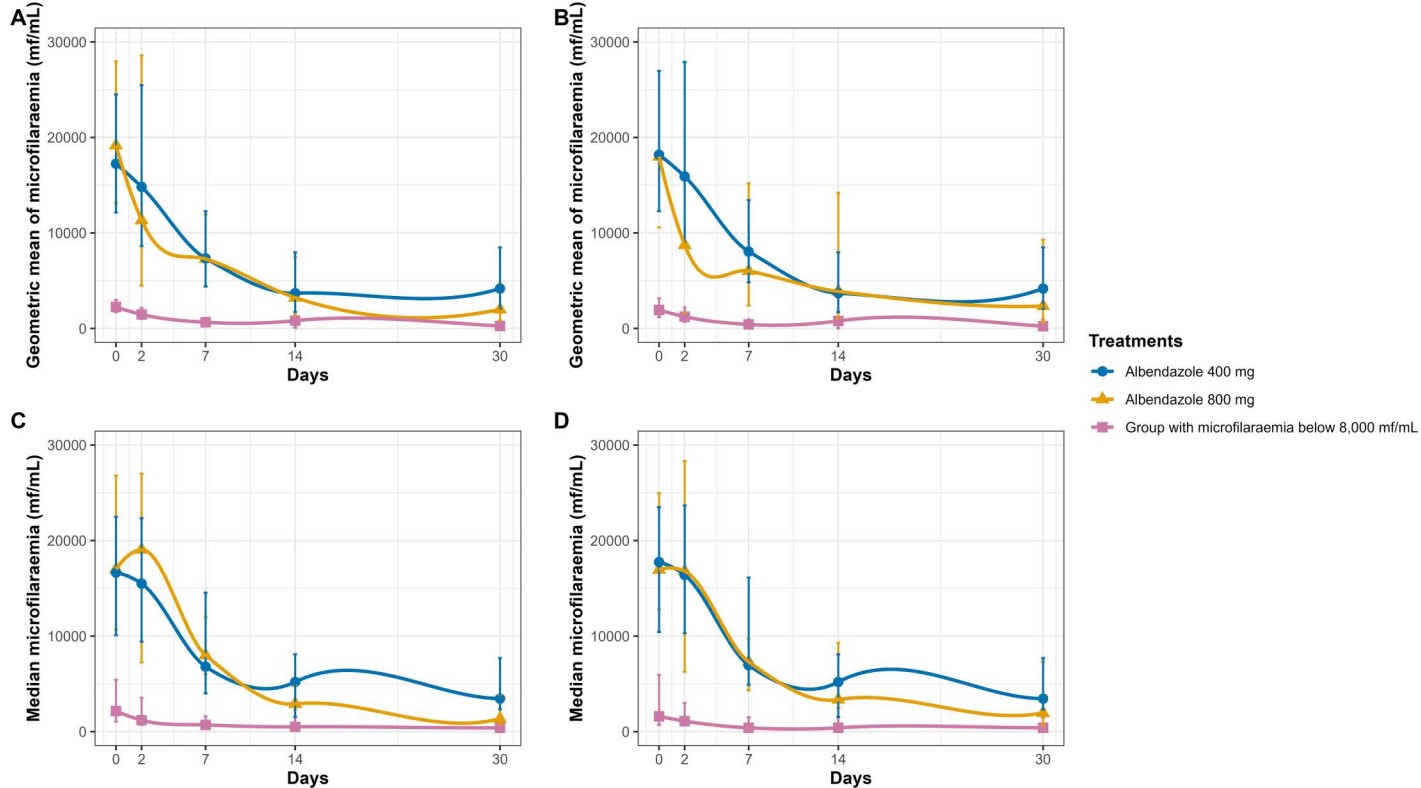

**Fig 4. Evolution of microfilaraemia in the treatment groups and the group with microfilaraemia below 8,000 mf/mL from Day 0 to Day 30.** Panels A and B show the intention-to-treat analysis, while Panels C and D present the per-protocol analysis. Panels A and C present the geometric mean of the microfilaraemia with the 95% confidence intervals while Panels B and D represent the medians with 25th and 75th percentiles. The microfilaraemia results presented correspond to the combination of direct examination of blood of 10 µL and leukoconcentration techniques. When direct examination is negative for *Loa loa*, the leukoconcentration method was subsequently performed.

**Table 3. Comparison of the geometric mean of the reduction rate (%) at each time point with D0.**

| | Intention-to-treat | | | Per-protocol | | |
|---|---|---|---|---|---|---|
| | Geometric mean of the reduction rate % (95% confidence interval) | | | Geometric mean of the reduction rate % (95% confidence interval) | | |
| Time points | ALB 400 mg | ALB 800 mg | p-value | ALB 400 mg | ALB 800 mg | p-value |
| D2 | 14.6 (-14.1 - 36.0) | 40.9 (-14.9 - 69.2) | 0.4 | 12.5 (-18.9 - 35.6) | 51.5 (-47.0 - 84.0) | 0.7 |
| D7 | 57.7 (39.0 - 70.7) | 63.4 (45.7 - 75.3) | 0.7 | 43.3 (35.1 - 69.8) | 54.5 (39.7 - 81.4) | 0.4 |
| D14 | 78.6 (62.0 - 89.1) | 80.6 (64.7 - 89.3) | 0.8 | 79.7 (62.0 - 89.1) | 80.4 (51.5 - 90.4) | 0.9 |
| D30 | 75.8 (60.7 - 86.6) | 89.6 (78.7 - 93.3) | 0.1 | 77.0 (60.7 - 86.6) | 86.9 (70.3 - 94.3) | 0.3 |

## Reduction in microfilaraemia below 8,000 mf/mL according to intention-to-treat and per-protocol analyses

In both treatment groups, the probability of remaining above or equal to 8,000 mf/mL decreased progressively over the 30-day of follow-up (Fig 5A and 5B). The probability decreased earlier in the group of patients with ALB 800 mg before 7 days of treatment contrary to the group receiving 400 mg (Fig 5A and 5B).

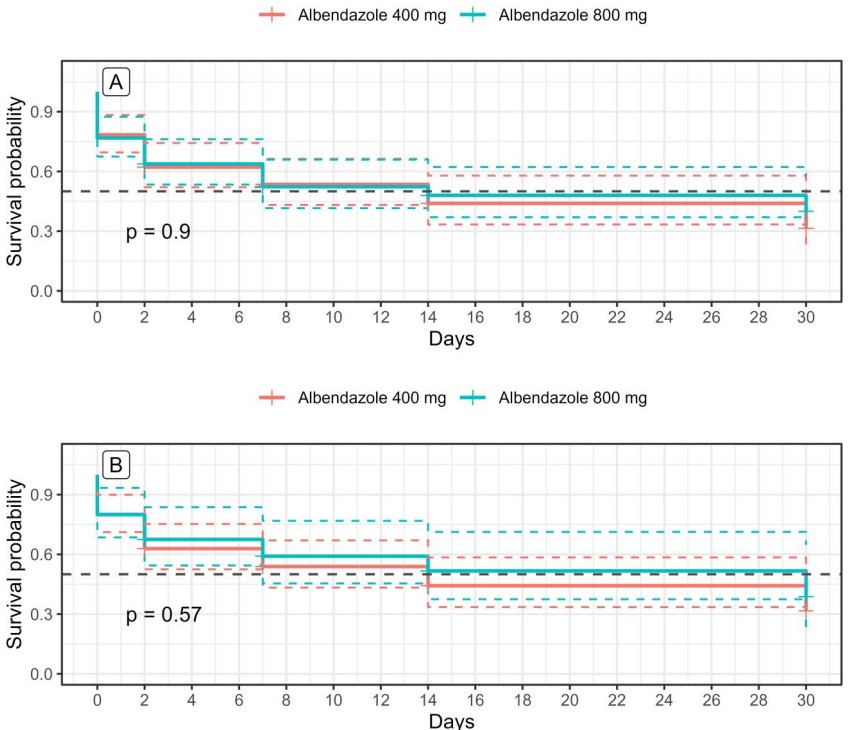

**Fig 5. Survival analysis.** This figure illustrates the proportion of hypermicrofilaraemic participants in the two treatment arms over time, in both the intention-to-treat (ITT) (A) and per-protocol (PP) (B) analyses. The dotted lines correspond to 95% confidence interval.

The probability that a patient would have microfilaraemia above or equal to 8,000 mf/mL decreased progressively over the 30-day follow-up in both treatment arms. On Day 2, this probability was estimated at 0.80 [0.62 - 1.0] for the 400 mg group and 0.68 [0.49 - 0.96] for the 800 mg group in the ITT analysis, while in the PP analysis, it was 0.86 [0.69 - 1.0] and 0.62 [0.36 - 1.0], respectively. These correspond to reductions of approximately 14.0-20.0% with 400 mg and 32.0-38.0% with 800 mg.

By Day 7, the probability further decreased to 0.40 [0.21 - 0.74] (400 mg) and 0.50 [0.31 - 0.81] (800 mg) in the ITT analysis, and 0.43 [0.23 - 0.78] and 0.37 [0.15 - 0.92] in the PP analysis - indicating reductions ranging from 50.0% to 63.0%.

At Day 14, probabilities declined further to 0.20 [0.07 - 0.55] and 0.17 [0.05 - 0.57] for the 400 mg and 800 mg groups, respectively, in the ITT analysis, and 0.21 [0.08 - 0.58] and 0.12 [0.02 - 0.78] in the PP analysis - equivalent to reductions of approximately 79% to 88%.

Finally, at Day 30, the probability of remaining above the 8,000 mf/mL threshold was 0.13 [0.09 - 0.48] in the ALB 400 mg group and 0.08 [0.01 - 0.53] in the ALB 800 mg group in the ITT analysis, while the PP analysis reported 0.14 [0.04 - 0.51] and 0.12 [0.02 - 0.78], respectively. These correspond to reductions of approximately 86% to 92%.

Despite these encouraging trends, no statistically significant difference was observed between the two dosages. In terms of patient distribution, half of the participants had microfilaraemia below 8,000 mf/mL by Day 7 in both treatment groups; however, the ITT analysis suggested this occurred slightly earlier for the 400 mg group (Day 7) and later for the 800 mg group (Day 11).

**PLOS Neglected Tropical Diseases**

## Evolution of symptoms

At baseline, 15.7% of patients (n = 11/70) presented with at least one symptom. The most common symptoms were ocular migration of a worm (26.7%, n = 6/11) and pruritus (24.1%, n = 5/11). The median number of symptoms was 2 [1–2] (Table 4).

In the group, 13.2% of patients (n = 5/38) exhibited at least one symptom, with the most frequent being sub-conjunctival migration of the adult worm, observed in 7.9% of patients. In the group receiving 400 mg of albendazole, 18.7% (n = 3/16) presented with symptoms, most notably sub-conjunctival migration, seen in 12.5% of patients. Among those treated with 800 mg of albendazole, 18.7% (n = 3/16) exhibited at least one symptom; the most common were pruritus and Calabar swelling (Table 4).

In all three groups, while most symptoms decreased over the course of treatment, pruritus initially increased before disappearing by Day 30 (Fig 6). With the exception of the group that received 400 mg of albendazole, in which 14.3% and 7.1% of patients reported pruritus and blurred vision respectively after 30 days of treatment, no symptoms were reported in any of the treatment groups at the end of the 30-day period (Fig 6).

## Incidence and progression of adverse events during treatment

A total of 11 (31.4%) patients in the group with microfilaraemia below 8,000 mf/mL, 6 (40.0%) patients in the group receiving 400 mg of albendazole, and 4 (28.6%) patients in the group receiving 800 mg of albendazole reported adverse events on Day 2 ($p = 0.78$). By the end of the treatment period, the frequencies of recorded adverse events had decreased, from 31.4% to 0.0% in the group with microfilaraemia below 8,000 mf/mL, from 40.0% to 14.3% in the 400 mg albendazole group, and from 28.6% to 16.7% in the 800 mg albendazole group. However, this decrease was statistically significant only in the group with microfilaraemia below 8,000 mf/mL, where no adverse events were reported after 30 days of treatment ($p < 0.01$) (Fig 7A).

On Day 2, in the group with microfilaraemia below 8,000 mf/mL, the most frequently reported adverse event was pruritus, observed in 14.3% of patients. Similarly, in the 400 mg albendazole group, pruritus was also the most common

**Table 4. Proportions of symptoms at the inclusion Day 0.**

| Symptoms | Albendazole 800 mg n (%) | Albendazole 400 mg n (%) | Group with microfilaraemia below 8,000 mf/mL n (%) |
|---|---|---|---|
| Calabar swelling[1] | 2 (12.5) | 0 (0.0) | 2 (5.3) |
| Eyeworm migration | 1 (6.3) | 2 (12.5) | 3 (7.9) |
| Pruritus | 2 (12.5) | 1 (6.3) | 2 (5.3) |
| Subcutaneous crawling | 1 (6.3) | 0 (0.0) | 1 (2.6) |
| Oedema[2] | 1 (6.3) | 0 (0.0) | 1 (2.6) |
| Blurred vision | 0 (0.0) | 1 (6.3) | 0 (0.0) |
| At least one symptom | 3 (18.7) | 3 (18.7) | 5 (13.2) |
| **Number of symptoms** | | | |
| 0 | 13 (81.3) | 13 (81.3) | 33 (86.8) |
| 1 | 1 (6.3) | 2 (12.5) | 2 (5.3) |
| 2 | 1 (6.3) | 1 (6.3) | 2 (5.3) |
| 3 | 0 (0.0) | 0 (0.0) | 1 (2.6) |
| 4 | 1 (6.3) | 0 (0.0) | 0 (0.0) |

Calabar swelling[1]: transient, localised, angioedema-type swelling caused by worm migration (specific to loiasis).

Oedema[2]: generalised or persistent swelling, often pitting, linked to systemic complications (non-specific).

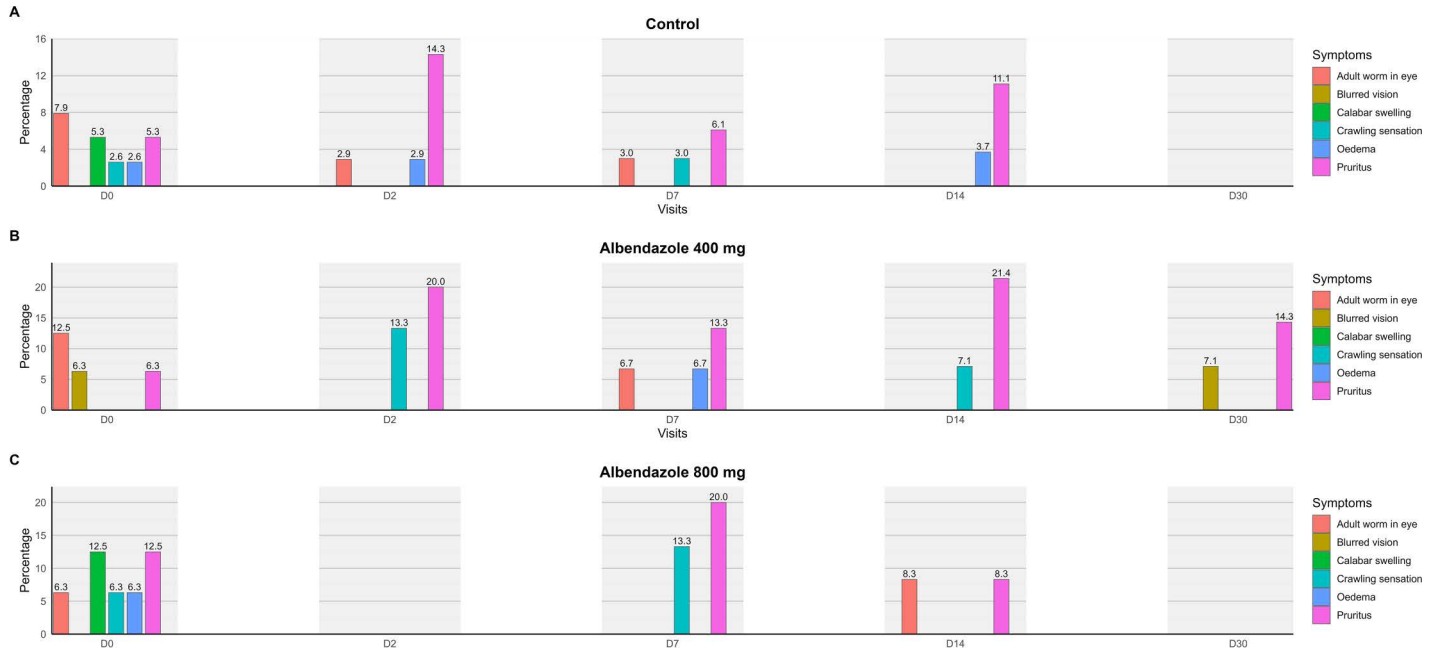

**Fig 6. Evolution of symptom frequency from Day 0 to Day 30 in the treatment groups and group with microfilaraemia below 8,000 mf/mL.**

symptom, reported by 20.0% of patients. In the 800 mg albendazole group, asthenia, fever, and drowsiness were the most frequently observed symptoms, each with a frequency of 7.1% (Fig 7B).

On Days 7 and 14, across all treatment groups, although some fluctuations occurred, the overall symptom profiles were comparable. Notably, a high frequency of fever was recorded in 70.4% of participants in the group with microfilaraemia below 8,000 mf/mL on Day 14.

By Day 30, with the exception of the group receiving 400 mg of albendazole - where 14.3%, 7.1%, and 7.1% of patients reported pruritus, increased appetite, and blurred vision respectively - no adverse events were recorded in the other two treatment groups (Fig 7C and 7D)

## Discussion

Elevated microfilaraemia (≥8,000 mf/mL) must be reduced prior to administering diethylcarbamazine (DEC) or ivermectin (IVM). Albendazole (ALB) is the drug of choice due to its low cost and its established use against other helminths, including gastrointestinal and blood parasites such as those responsible for lymphatic filariasis. ALB allows for a gradual reduction in microfilaraemia before initiating macrofilaricidal therapy, facilitating patient recovery [15]. Although therapeutic protocols have been developed, they have not yet been validated using methodologies recognised in experimental drug development within clinical research, i.e., methods accepted by the scientific community and necessary for standardisation and deployment in all loiasis-endemic regions [16]. Given the co-endemicity of loiasis and onchocerciasis in areas such as Gabon, these protocols could render individuals in endemic areas eligible for the eradication of this neglected tropical disease. Furthermore, recent studies suggest that hypermicrofilaraemia may be associated with higher mortality rates, particularly among young adults [17,18].

Treatment of loiasis must be effective across different strata of microfilaraemia: low, moderate, and high. We recently reported data from a clinical trial evaluating ALB *versus* IVM in patients with low microfilaraemia ranging from 500 to 3,500 mf/mL [11]. Another study conducted in Lambaréné assessed patients with microfilaraemia between 5,000 and 50,000 mf/mL

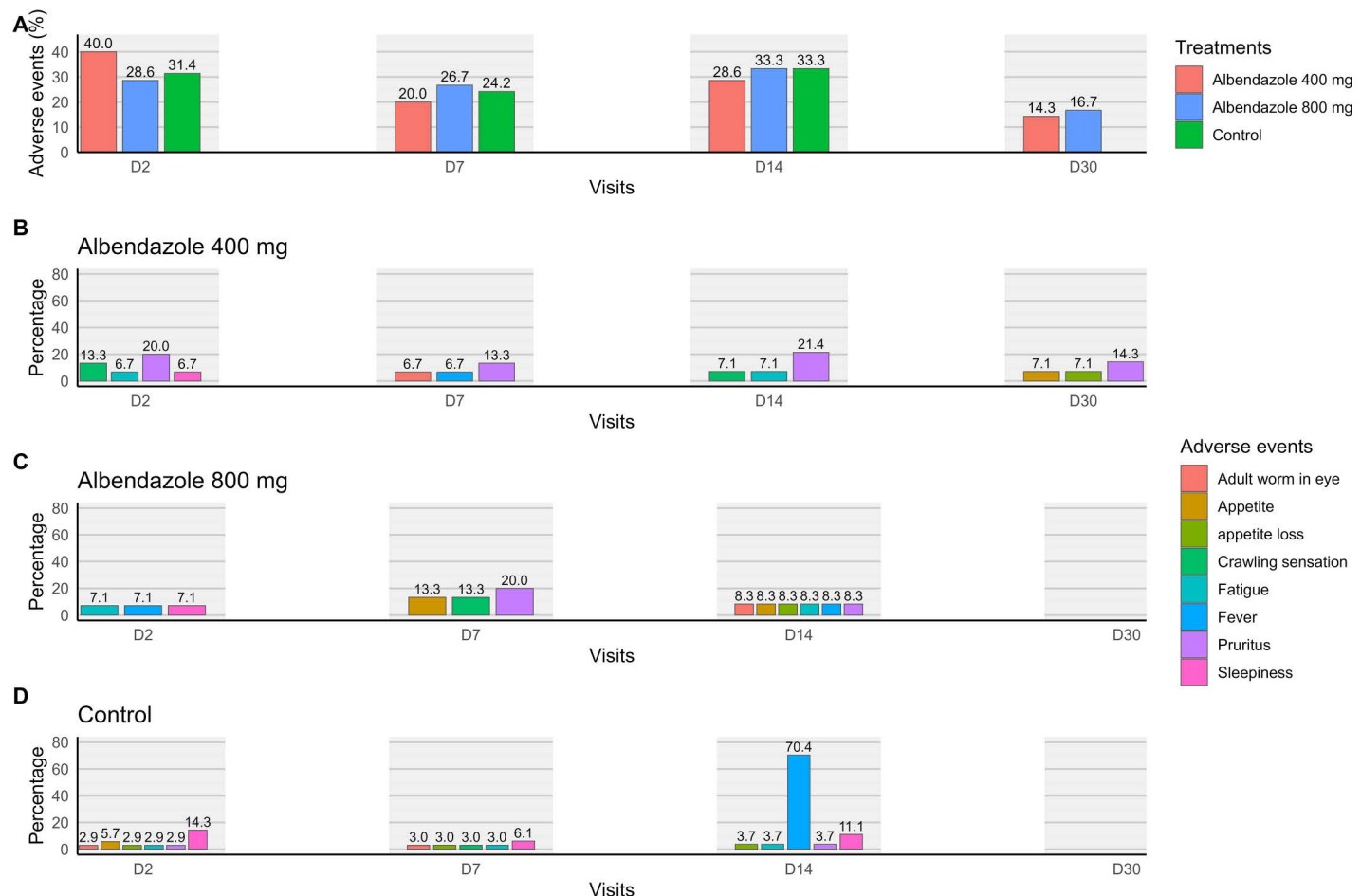

**Fig 7. Evolution of adverse event frequency in the treatment groups and the group with microfilaraemia below 8,000 mf/mL throughout the follow-up period.**

treated with ALB over 35 days [19]. Additional data from various settings in Gabon are needed to address the critical gap in the adoption of a standardised national protocol, not only for the pre-treatment of hypermicrofilaraemic loiasis but also for the routine management of infected patients.

In our group with microfilaraemia below 8,000 mf/mL (treated with ALB 400 mg daily for 30 days), the reduction in microfilarial density between Day 0 and Day 14 was already substantial, with median reductions of around 50.0%. By contrast, in the double-blind placebo-controlled trial conducted by Klion et al. in Benin, where patients received albendazole 400 mg/day (200 mg twice daily for 21 days), microfilarial densities remained largely unchanged until Day 14, at which point the first significant decreases compared with placebo were observed. This temporal difference suggests that, in our study, parasite clearance under albendazole occurred more rapidly than in the Beninese cohort. Several factors may explain this discrepancy, including possible variations in study design and treatment supervision.

In addition to methodological differences, interindividual and population-level pharmacogenetic variability may help explain the discrepancies reported across studies. Variations in cytochrome P450 enzyme activity (particularly within the CYP2J2 and CYP2C19 subfamilies involved in albendazole hydroxylation) can influence drug bioavailability, metabolism, and ultimately therapeutic efficacy [20]. These enzymatic activities depend not only on genetic polymorphisms but also on

environmental and behavioural factors, including alcohol and tobacco use, which are known to modulate CYP450 isoenzymes. Recent data from Gabon show that alcohol and tobacco consumption are more common in rural than in urban settings (19.4% vs 9.6% and 26.1% vs 6.2%, respectively), especially among men, which may alter hepatic metabolism and affect albendazole pharmacokinetics [21]. Differences in the genetic diversity of *Loa loa* populations across Central Africa (notably in β-tubulin genes, the molecular target of benzimidazoles) may also influence parasite susceptibility to treatment. Taken together, these metabolic, behavioural, and genetic factors from both host and parasite may help account for the heterogeneity observed among clinical trials conducted in various endemic regions, including our own.

Consistent with the findings of Klion et al. [7], our data indicate that albendazole administered daily for 30 days can substantially lower microfilarial densities and thereby reduce the number of individuals who would otherwise be excluded from ivermectin treatment under a Test and Treat strategy. However, as in the Beninese trial, the reductions observed in our study were not sufficient to bring all patients below the 8,000 mf/mL safety threshold. Consequently, individual microfilarial quantification remains indispensable before treatment, and this approach cannot be used as a substitute for mass drug administration without prior screening.

This randomised, controlled, single-blind Phase IIb clinical trial was conducted to evaluate patients with microfilaraemia above the 8,000 mf/mL threshold. The main findings indicate that hypermicrofilaraemic patients treated with ALB 400 mg and 800 mg for 30 days achieved microfilaraemia reductions below 8,000 mf/mL in 71.4% and 83.3% of cases, respectively, in the intention-to-treat (ITT) analysis, and in 71.4% and 75.0%, respectively, in the per-protocol (PP) analysis. Although the percentage was higher with the 800 mg dose, the difference was not statistically significant. This suggests that increasing the dose may not confer additional benefit in reducing microfilaraemia below the critical threshold of 8,000 mf/mL. Importantly, our analyses also showed that microfilarial densities decreased more rapidly in the 800 mg group than in the 400 mg group. While this accelerated decline may raise safety concerns in hypermicrofilaraemic individuals, particularly with respect to the risk of adverse events. Regarding patient adherence, it is preferable to administer a single tablet rather than two, as supported by literature on compliance with chronic disease treatments [22].

To date, no other protocols in the literature match the approach used in this Phase IIb trial. However, some previous studies have investigated similar regimens. For example, Tsague-Dongmo and colleagues developed a treatment based on ALB 800 mg administered over three days for patients with microfilaraemia ≥ 8,000 mf/mL. A progressive reduction in microfilaraemia was observed over the first three months, although no significant difference was reported [23]. This supports our findings, indicating that albendazole may induce a gradual decrease in microfilaraemia.

In another study conducted in Gabon, hypermicrofilaraemic patients were treated with ALB 800 mg, administered in two doses of 400 mg over 35 days. This trial reported a significant 91.0% reduction in microfilaraemia (95% CI: 44.0-99.0) after 30 days [19]. Although this reduction was approximately 8% higher than that observed in our study, the difference may be attributed to variations in inclusion criteria, particularly regarding microfilaraemia levels. In that study, the median microfilaraemia was 9,225 mf/mL (7,300–16,025), with patients included having levels from 5,000 to 50,000 mf/mL, roughly half of the median parasitaemia detected in our study (17,000 mf/mL [8,200–105,900]), where only patients with ≥ 8,000 mf/mL were included [19]. Despite these methodological differences, the findings of Zoleko-Manego and colleagues align with ours, demonstrating an overall downward trend in microfilaraemia. Indeed, in these various trials, a significant reduction in microfilaraemia below 8,000 mf/mL was observed before Day 7 of treatment [19]. This underscores the efficacy of an 800 mg albendazole regimen in reducing microfilarial loads in patients with high parasite burdens, who are at increased risk of severe adverse events if treated with diethylcarbamazine (DEC) or ivermectin (IVM). Our own results confirm that the decline occurred earlier and more rapidly in the 800 mg group than in the 400 mg group. While this accelerated clearance reflects the strong efficacy of high-dose albendazole, it may also represent a safety concern in hypermicrofilaraemic individuals, who are at greater risk of adverse reactions when parasite loads fall abruptly. This observation emphasises the need to carefully weigh efficacy

against safety when considering higher doses of albendazole. However, Zoleko-Manego and colleagues did not include a 400 mg albendazole arm. Our results show no significant difference between the 800 mg and 400 mg groups when comparing the reduction rate of microfilaraemia between D0 and D30, suggesting that increasing the dose may not confer additional benefit in reducing microfilaraemia below 8,000 mf/mL. These findings may inform optimisation of treatment protocols by balancing efficacy with the drug burden.

In the ALB 400 mg group, it was observed a slight increase in microfilaraemia between Day 14 and Day 30, despite the overall downward trend. A similar pattern was reported by Zoleko-Manego and colleagues, in 2023, in their Gabonese cohort, suggesting that albendazole may not fully suppress repopulation of microfilariae once treatment ceases. This rebound could reflect the persistence of viable adult worms and highlights the need for follow-up studies assessing the durability of albendazole's effects and the potential requirement for combination or repeated therapy.

Our findings are also consistent with those of Klion et al. [7], who conducted a double-blind placebo-controlled Phase IIb trial in Benin using albendazole 400 mg twice daily for 21 days. In that study, as in ours, some but not all patients with high microfilarial densities experienced reductions below the 8,000 mf/mL threshold. These converging results indicate that albendazole, while not uniformly effective in clearing microfilariae, can nevertheless lower microfilarial loads sufficiently to reduce the number of individuals excluded from further treatment. However, they also reinforce the necessity of microfilarial quantification before treatment, since a significant proportion of patients remain above risk thresholds after therapy.

Regarding adverse events, 40.0% of patients in the 400 mg group and 28.6% in the 800 mg group reported adverse events after two days of treatment. One might anticipate a higher rate in this study compared to that of Zoleko-Manego and colleagues, given the high median microfilaraemia levels in both populations. Interestingly, the adverse event rates were relatively similar, suggesting that higher microfilarial loads are not necessarily associated with increased frequency or severity of adverse events during albendazole therapy. While the current study, alongside that of Zoleko-Manego et al., supports the safety of albendazole in populations with heavy parasite burdens, there has been at least one report of a serious adverse event following treatment.

Likewise, Volpicelli et al. published a case of albendazole-associated encephalopathy during a 21-day treatment course, after exclusion of alternative causes, thereby further supporting a causal relationship and implicating host-parasite inflammatory interactions as a potential mechanism [24].

Taken together, these cases underline that albendazole, while promising as a preparatory or alternative therapy, must be used within carefully validated protocols, especially in individuals with high parasite loads. Although our trial and that of Zoleko-Manego et al. [19] support the relative safety and tolerability of prolonged albendazole in populations with heavy parasite burdens, the occurrence of significant but rare severe neurological complications, cytopenias and Stevens-Johnson syndrome in the literature demonstrate that risk cannot be excluded [25–27]. These findings strengthen the rationale for cautious dose optimisation, rigorous monitoring, and the development of clear guidelines for the management of loiasis. In this context, our results suggest that the 400 mg once-daily regimen may represent the best compromise, offering similar efficacy to the 800 mg regimen while potentially minimising the risk of abrupt parasite clearance and associated adverse events.

Overall, all experimental regimens were found to be effective and well tolerated in this clinical trial. Furthermore, no significant difference was observed between the two intervention groups, despite one receiving double the dose. This suggests that the 400 mg once-daily regimen is optimal, demonstrating similar efficacy and safety to the 800 mg regimen (administered as two 400 mg tablets). However, the rate of microfilarial decline was clearly faster in the 800 mg group, which could translate into a higher risk of adverse events in hypermicrofilaraemic subjects. This suggests that while the 400 mg once-daily regimen is optimal for balancing efficacy and safety, the 800 mg regimen should be used with caution and warrants further evaluation in larger safety-focused studies. This finding is particularly important as it may reduce drug pressure while maintaining therapeutic efficacy. Additionally, dose reduction could offer clinical and practical advantages.

A lower dose may minimise potential side effects in case of very high microfilaraemia and improve adherence, while also reducing treatment costs, potentially enhancing access in rural, low-resource settings.

It should be noted that the study population was relatively old, with a mean age over 45 years across treatment groups. This is likely due to the predominance of older individuals in rural Gabonese villages and the known increase in loiasis prevalence with age in these settings [11]. Socio-economic and environmental factors, such as migration of young adults to urban areas for work or education, may also contribute to this trend.

Our study fits within this context by evaluating the efficacy and safety of albendazole at 400 mg, a drug already widely used for helminthiases, but whose potential in reducing microfilarial loads in hypermicrofilaraemic patients is not yet fully exploited. Providing data on the minimal effective dose contributes to the optimisation of therapeutic protocols, facilitating their deployment in rural resource-limited settings where loiasis control remains a public health priority. The findings of this study have significant public health implications, particularly in establishing an effective and affordable pre-treatment strategy to reduce microfilaraemia prior to administering high-risk treatments such as diethylcarbamazine (DEC) or ivermectin (IVM). The simplicity of administering the 400 mg once-daily regimen could facilitate its adoption within national loiasis control programmes, especially in rural settings where healthcare access is limited. Furthermore, this approach has the potential to enhance the safety profile of mass drug administration campaigns by decreasing the incidence of severe adverse reactions, thereby increasing prevention coverage, a critical factor in achieving disease elimination.

The main limitation of this study was the absence of follow-up over longer durations to assess the persistence of therapeutic effects, since it is known that complete parasite clearance cannot be observed after only one month of treatment; effective clearance typically occurs at 3 or 6 months with additional molecules. Equally, the impracticality of a 30-day albendazole regimen, which may reduce adherence and increase the risk of non-compliance under real-world conditions. To finish, no biochemical laboratory testing was performed during the trial knowing that albendazole can cause hepatotoxicity and bone marrow suppression in some patients at the dose used in this study.

For a phase IIb clinical trial, the relationship between microfilaraemia and albendazole metabolite concentrations (pharmacokinetics) should also be evaluated.

Further studies are needed to confirm the long-term sustainability of the therapeutic effects, including follow-up over 6–12 months. The pharmacokinetics of albendazole in this specific continuously-exposed population should also be better characterised to optimise dosing and minimise the risk of rare adverse events, particularly neurological complications. Additionally, reproducibility of these results across different epidemiological and geographic contexts should be evaluated to facilitate wider implementation.

## Conclusion

This study demonstrates that albendazole at 400 mg once daily for 30 days is as effective and well tolerated as the 800 mg regimen in patients with hypermicrofilaraemia. The comparable efficacy, combined with the potential benefits of lower drug burden, reduced costs, and improved adherence, supports the adoption of the 400 mg regimen as the optimal protocol in endemic settings while limiting potential adverse effects. These findings contribute valuable evidence towards the development of standardised, validated pre-treatment strategies that can safely reduce microfilarial loads, facilitating the safe administration of ivermectin for MDA or diethylcarbamazine for treatment. Implementing such protocols could significantly advance efforts to control and eliminate loiasis in endemic regions, ultimately reducing disease burden and associated morbidity.

## Acknowledgments

The authors are grateful to all the staff at the Centre de Recherche biomédicale En pathogènes Infectieux et Pathologies Associées (CREIPA) and the Unité Mixte de Recherche sur les Agents Infectieux et leur Pathologie (UMRAIP) for their support in participant recruitment in Bitam, and for their assistance with sample processing in both Libreville and Bitam. Special thanks are also to Bifolossi Medical Center staff for hosting our study. We extend our gratitude to the

village chiefs and other relevant authorities for their support throughout this research. Espediee CHABI, Junior Dimitri MOUDOUMBI-MOUANDZA, Stéphane OGOULA, Ivan-Guy-Tiburce NDAMA-NDAMA, Lys Anna HOMBO NDOMANA, Melly Chancelle MIMBILA MINKO, Ndjena Andress OBALI, Télesphore OBIANG NGOUA and Rochat Léotard SIMA OWONO for their work on the field and in the laboratory. Finally, we thank all the participants whose cooperation made this study possible.

## Author contributions

**Conceptualization:** Noé Patrick M'Bondoukwé, Marielle Karine Bouyou Akotet.

**Data curation:** Noé Patrick M'Bondoukwé, Luccheri Ndong Akomezoghe, Luice Aurtin Joel James.

**Formal analysis:** Noé Patrick M'Bondoukwé, Luccheri Ndong Akomezoghe, Ginette Severine Zang Ondo, Luice Aurtin Joel James.

**Funding acquisition:** Noé Patrick M'Bondoukwé, Marielle Karine Bouyou Akotet.

**Investigation:** Noé Patrick M'Bondoukwé, Luccheri Ndong Akomezoghe, Bridy Chesly Moutombi Ditombi, Jacques Mari Ndong Ngomo, Hadry Roger Sibi Matotou, Valentin Migueba, Coella Joyce Mihindou, Bedrich Pongui Ngondza, Christian Mayandza, Reinne Moutongo.

**Methodology:** Noé Patrick M'Bondoukwé, Luccheri Ndong Akomezoghe, Denise Patricia Mawili Mboumba.

**Project administration:** Dimitri Ardin Moussavou Mabicka, Denise Patricia Mawili Mboumba, Marielle Karine Bouyou Akotet.

**Resources:** Marielle Karine Bouyou Akotet.

**Supervision:** Denise Patricia Mawili Mboumba, Marielle Karine Bouyou Akotet.

**Validation:** Charleine Manomba, Denise Patricia Mawili Mboumba, Marielle Karine Bouyou Akotet.

**Visualization:** Noé Patrick M'Bondoukwé, Luccheri Ndong Akomezoghe, Luice Aurtin Joel James.

**Writing – original draft:** Noé Patrick M'Bondoukwé, Luccheri Ndong Akomezoghe, Ginette Severine Zang Ondo.

**Writing – review & editing:** Bridy Chesly Moutombi Ditombi, Denise Patricia Mawili Mboumba, Marielle Karine Bouyou Akotet.

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
