## [Decision Letter · Decision Letter 0]

30 Jun 2025

Assessment of the Efficacy and Safety of Two Albendazole Regimens for the treatment of hypermicrofilaraemic loiasis in adult populations in Woleu-Ntem Province, Gabon: a Phase IIb single-blind randomised controlled trial

Dear Dr. M'Bondoukwé,

Thank you for submitting your manuscript to PLOS Neglected Tropical Diseases. After careful consideration, we feel that it has merit but does not fully meet PLOS Neglected Tropical Diseases's publication criteria as it currently stands. Therefore, we invite you to submit a revised version of the manuscript that addresses the points raised during the review process.

Please submit your revised manuscript within 60 days Aug 29 2025 11:59PM. If you will need more time than this to complete your revisions, please reply to this message or contact the journal office at plosntds@plos.org. Please include the following items when submitting your revised manuscript:

We look forward to receiving your revised manuscript.

Kind regards,

Siddhartha Mahanty, M.B.B.S., M.P.H

Academic Editor

Jong-Yil Chai

Section Editor

Shaden Kamhawi

co-Editor-in-Chief

Paul Brindley

co-Editor-in-Chief

**Additional Editor Comments:**

Thank you for submitting your manuscript which addresses a neglected but major public health issue in Loa loa-endemic regions of Africa. All the reviewers acknowledged the significance of the study for the question addressed (management of hyper microfilaraemia in Loa loa-endemic regions) and the size of the study. However, there are significant issues with the study design and reporting of results that have been detailed by the reviewers, particularly Reviewers 1 and 2. Please address all the issues raised, paying particular attention to the questions raised by Reviewers 1 and 2 about selection of a control group and levels of microfilaraemia for inclusion in the study groups. We will consider a revised manuscript for publication after re-review if you are able to adequately address the reviewers concerns.

**Journal Requirements:**

At this stage, the following Authors/Authors require contributions: Noé Patrick M'Bondoukwé, Luccheri Ndong Akomezoghe, Bridy Chesli Moutombi Ditombi, Jacques Mari Ndong Ngomo, Hadry Roger Sibi Matotou, Ginette Sévérine Zang Ondo, Valentin Migueba, Coella Joyce Mihindou, Bedrich Pongui Ngondza, Christian Mayandza, Héléna Kono, Dimitri Hardrin Moussavou Mabicka, Charleine Manomba, Reinne Moutongo, Luice Aurtin Joel James, Denise Patricia Mawili Mboumba, and Marielle Karine Bouyou Akotet. Please ensure that the full contributions of each author are acknowledged in the "Add/Edit/Remove Authors" section of our submission form.

- ® on page: 8.

Potential Copyright Issues:

i) Figure 1. Please (a) provide a direct link to the base layer of the map (i.e., the country or region border shape) and ensure this is also included in the figure legend; and (b) provide a link to the terms of use / license information for the base layer image or shapefile. We cannot publish proprietary or copyrighted maps (e.g. Google Maps, Mapquest) and the terms of use for your map base layer must be compatible with our CC BY 4.0 license.

6) We note that your Data Availability Statement is currently as follows: "N/A". Please confirm at this time whether or not your submission contains all raw data required to replicate the results of your study. Authors must share the “minimal data set” for their submission. PLOS defines the minimal data set to consist of the data required to replicate all study findings reported in the article, as well as related metadata and methods (https://journals.plos.org/plosone/s/data-availability#loc-minimal-data-set-definition).

7) Please amend your detailed Financial Disclosure statement. This is published with the article. It must therefore be completed in full sentences and contain the exact wording you wish to be published.

2) If any authors received a salary from any of your funders, please state which authors and which funders..

**Reviewers' Comments:**

Reviewer's Responses to Questions

**Key Review Criteria Required for Acceptance?**

**Methods**

-Are the objectives of the study clearly articulated with a clear testable hypothesis stated?

-Is the study design appropriate to address the stated objectives?

-Is the population clearly described and appropriate for the hypothesis being tested?

-Is the sample size sufficient to ensure adequate power to address the hypothesis being tested?

-Were correct statistical analysis used to support conclusions?

-Are there concerns about ethical or regulatory requirements being met?

Reviewer #1: The trial includes three groups of individuals: two groups of subjects whose L. loa microfilaraemia was ≥8000 microfilariae per mL (one treated with 400 mg albendazole daily for 30 days, the other treated with 800 mg albendazole daily for 30 days); and a so-called control group, including individuals with a microfilaraemia <8000 mf/mL who received 400 mg albendazole daily for 30 days. Can we really consider this third group as a "control group"? Why didn't the authors use a placebo group in their trial?

The methods used to assess the microfilaraemia at the various stages of the trial should be clarified. In the sub-section called "Data collection and screening", it is said that "a 4 mL venous blood sample was collected in an EDTA tube for parasitological screening of L. loa via direct examination of 10 μL of blood". Was it necessary to collect venous blood at this stage of screening ? Why didn't the authors examine a larger volume of blood (50 µL) taken from the fingertip either by direct examination of the wet smear, or by examination of stained thick smears? Was the leucoconcentration method systematically used during this screening stage for all the individuals?

The most interesting part of this trial is the comparison of the effects of 400 and 800 mg regimens in the hypermicrofilaraemic subjects. For each timepoint (D2, D7, D14 and D30), the authors should compare the distribution of the individual reduction rates (from D0), using nonparametric Wilcoxon rank-sum tests (Mann-Whitney U test).

Reviewer #2: In the introduction, the authors mention the need to find a way to decrease Loa loa microfilaremia in individuals who fail Test and Treat. However, the threshold used to exclude individuals from ivermectin treatment in the Test and Treat studies was 20,000 mf/mL. Why did the authors select 8,000 mf/mL as the threshold for their study since DEC is not used in areas endemic for onchocerciasis. At a minimum, analyses should be included using the 20,000 mf/mL cutoff.

The reason for the particular choice of control group needs to be made more clear. Presumably, the use of a placebo-control was not acceptable to the population. However, the added benefit of the control group is unclear. An untreated group of patients with high microfilaremia would have been more appropriate.

Why was cetirizine given for 7 days to the control group and albendazole 800 mg? This is not standard concomitant therapy for albendazole, and the antiparasitic effect of albendazole in loiasis takes weeks (as supported by the fact that the patients in this study developed pruritus after 30 days of treatment). Was this also given to the albendazole 400 mg treatment group and, if not, why not, as this would preclude comparison between the groups?

Reviewer #3: yes

**Results**

-Does the analysis presented match the analysis plan?

-Are the results clearly and completely presented?

-Are the figures (Tables, Images) of sufficient quality for clarity?

Reviewer #1: The results of the nonparametric tests should be added in the Results subsection called "Reduction in Microfilaraemia in Intention-to-Treat and Per-Protocol Analyses". Figures 3A and 3B should be completed with new Figures 3C and 3D showing the changes in the MEDIANS (and interquartile ranges) of the microfilarial densities at each timepoint.

Reviewer #2: The actual time points are not provided in the text or Figure 2. There are time points in the legends to Figure 5 and 6 but the bars are distributed across the axis, so it is not entirely clear what the actual time point and window were. A figure showing the study design would help. Were there safety of efficacy time points after the 30 day treatment? As the authors note, the effect of albendazole on microfilaremia typically peaks at 6 weeks and continues for months.

Were any of the participants co-infected with other filariae (Onchocerca, Mansonella,..)?

What were the reasons for exclusion of 84% (336/400) of the Loa-positive patients? This seems like a very large number given the exclusion criteria. What is meant by “unvisited” in Figure 2 – again these numbers are significant (25% of the patients in the high dose group were unvisited at the final time point).

What is meant by eosinophilia rate in % in Table 1 – the percent of people with eosinophilia or the mean eosinophil % in each group? Assuming the latter, % eosinophil count is not very useful as it can vary substantially depending on the neutrophil count (absolute eosinophil count is a better reflection of eosinophilia). That said, % eosinophil count is not normally distributed and should be provided as geometric mean and range. Would consider adding range to the geometric mean mf/mL as well.

In Figure 4, the authors appear to be showing the proportion of participants meeting the 8,000 mf/mL threshold. However, the text talks about the geometric means and probabilities. This is very confusing. In any case, this data is consistent with prior studies of albendazole, including a placebo-controlled study of albendazole 400 mg twice daily for 21 days, which have shown that some but not all patients with high microfilarial levels show decreases below 8,000 mf/mL. Of note, this earlier trial is mentioned in the introduction (ref 7) where it is incorrectly cited to suggest that albendazole acts on microfilariae but is not mentioned in the discussion despite the fact that the regimen used was most closely aligned with the current paper and it had a placebo-controlled phase IIb design.

Reviewer #3: yes

**Conclusions**

-Are the conclusions supported by the data presented?

-Are the limitations of analysis clearly described?

-Do the authors discuss how these data can be helpful to advance our understanding of the topic under study?

-Is public health relevance addressed?

Reviewer #1: In the third paragraph of the discussion, the authors simply compare the percentage of people who fell below the 8,000 mf/mL threshold between the two hypermicrofilaraemic groups. Actually, one of the main result of the study is that the microfilarial densities decreased much more rapidly in the group treated with 800 mg than in those treated with 400 mg. This poses a major safety problem. A regimen using 800 mg is probably not appropriate to treat hypermicrofilaraemic subjects because it could lead to more frequent serious adverse events. The text in paragraphs 5 and 8 of the discussion, as well as the abstract, should be revised accordingly. In addition, the authors should compare the decrease in microfilarial density they observed in their control group between D0 and D14 and the decrease shown by Klion et al. (reference 7) in subjects from Benin who had received the same dose of albendazole (400 mg per day). The decrease seen by Klion et al. seems to be more progressive. What could be the reasons for this?

In paragraph 6 of the discussion, the authors mention the case of post-albendazole encephalopathy described by Métais et al. They could have also mentioned the case reported by Volpicelli et al. (Parasitology International 2020).

In paragraph 7 of the discussion, the authors state that "their findings contrast with those of Gobbi and colleagues, in whom 50.0% of patients treated with 400 mg/day of albendazole remained symptomatic after 30 days". This is not very relevant because among those 5 subjects treated with albendazole only in the Gobbi et al. paper, only two were followed up clinically.

Reviewer #2: The impracticality and potential for non-compliance of administering 30 days of albendazole need to be addressed in the discussion as this is a limitation to the approach suggested.

Albendazole is known to cause hepatotoxicity and bone marrow suppression in some patients at the doses used in this study. Since laboratory testing was not performed either prior to or during the trial, this limitation also needs to be addressed.

As concluded in the double-blind placebo controlled trial of albendazole in loiasis (reference 7), the data form the current study indicate that albendazole given for up to a month would potentially reduce the number of people left untreated during a Test and Treat campaign but would not allow MDA without microfilarial quantification.

Reviewer #3: yes

**Editorial and Data Presentation Modifications?**

Reviewer #1: - Introduction, paragraph 1, line 1, please replace "Loa (L.) loa" by "Loa loa"

- Introduction, para 1, line 7 : I am not sure that references 2 and 3 are appropriate there

- Introduction, para 1, lines 5-6: "in individuals … fatal outcomes". Please provide the reference of the papers where the numbers (200-fold, 1,000-fold) come from, as well as information on the reference group. Does this group correspond to the amicrofilaraemic individuals?

- Introduction, para 1, last line: please spell out PNLMT

- Introduction, para 2: please explain what the "alternative non-pharmacological strategies" could be

- Introduction, para 2, line 4. By definition, a test and treat strategy is not a mass treatment. Thus replace "mass" by "large-scale"

- Introduction, para 2, last line. I am not sure reference 4 is appropriate there

- Introduction, para 3, line 5. Please replace reference 5 by another reference giving results on the impact of biannual treatment with albendazole alone on the prevalence of lymphatic filariasis

- Introduction, para 3, line 8: are the authors sure that albendazole "acts on microfilariae"? Please explain what it means (microfilaricidal?, embryostatic?, embryotoxic?) and provide a reference

- Introduction, para 3, line 9: please delete "Recently" (the paper (reference 4) has been published in 2016)

- Introduction, para 3, lines 11-12: what is the controversy?

- Introduction, para 4, lines 3-4: what is the meaning of "to use effective molecules for parasitic diseases in the same geographical areas"? Does this refer to integration of NTD control activities? Please clarify

- Introduction, para 4, line 4, please replace "The DPMTM" by "The Department of Parasitology-Mycology-Tropical Medicine (DPMTM) of the Libreville Faculty of Medicine". And delete the details at the first line of the Discussion section

- Introduction, para 4, penultimate line: reference 11 is a preprint posted in August 2024. Why do the authors mention PLoS NTD in the reference?

- Methods, Inclusion criteria: replace "weigh" by "weight"

- Methods, Exclusions criteria: what was the interval of time between the selection procedure and the first dose of albendazole?

- Methods, Study design, Study drugs: add "of a village community worker" after "under direct supervision" ; and delete reference to the village community workers in the next paragraphs

- Methods, Study design, Study drugs: please replace "after a fatty meal" by "15 to 30 minutes after a fatty meal (such as fatty rolls with approximately 15 g of butter)". And delete the information on the fatty meal in the "Patients follow-up" sub-section

- Methods, Study design, Study drugs: if all the groups have received a course of 10 mg cetirizine tablet daily for 7 days, this could be mentioned either before, or after the three sub-sections describing the albendazole regimens, to avoid redundancy

- Methods, Study design, Safety parameters: please replace "or ocular circulation related to" by "or subconjunctival migration of"; and clarify the difference between "Calabar swelling" and "oedema" (the two conditions are distinguished in Table 2)

- Methods, Study design, Patients follow-up : please mention in this sub-section that microfilaraemia was reassessed on D2, D7, D14 and D30. Please explain whether, once the individuals have been selected for inclusion in the trial (after the screening stage), their microfilaraemia was again measured just before the first albendazole treatment, providing the value at D0. Please clarify whether efforts were made to examine the participants at the same time (hour, minute) of the day at each follow-up visit, in order to limit the impact of the diurnal periodicity of microfilaraemia. The footnote of Figure 2 (No participants were lost to follow-up, as the study design allowed visits at two non-consecutive time points) is not very clear.

- Methods, Statistical analysis: please mention here that the analyses were conducted using both ITT and PP approaches. And see major comment on the use of nonparametric tests to compare reduction rates

- Results, table 1 : in the second line, please explain whether the numbers and percentages correspond to males or to females

- Results, Figure 4 : please explain what the dotted lines correspond to. Do they show the 95% confidence intervals?

- Results, Figure 5, 6B, 6C and 6D: it would be great if the authors could add vertical lines to separate the groups of histograms at each day of follow-up

- Discussion, para 1, line 10: I am not sure that reference 16 is relevant there

- Discussion, para 1, lines 10-12: what the authors mean by "these protocols could render the entire population in endemic areas eligible for the eradication of this neglected tropical disease"? Do the authors think that the albendazole courses could be used on a large scale, as they suggest in paragraph 10 of the discussion and in the conclusion? Or would they be applicable only at individual level?

- Discussion, para 4: please clarify "no significant difference". When compared to the value at D0?

- Discussion, para 9, line 1 : replace "older" by "old" or explain what is the comparison population

Reviewer #2: Exclusion criteria are labelled as “non-inclusion”. Reasons for withdrawal are labelled as “exclusion”. This is confusing.

Figure 2 is also confusing. The time points are not provided. “Unvisited participants” are listed and it is stated that no patients were withdrawn from study but it is unclear whether any of the participants missed multiple time points and who was included in the PP vs. ITT analyses.

Per the text, Figure 3 shows geometric mean reduction in the three groups. A measure of the data dispersion at each point should be added both to the text and the figure. The figure legend should also indicate what is being shown (i.e., geometric mean).

Red and green should not be used as the two colors in the graphs in Figure 4 as approximately 10% of males are red-green colorblind.

Reviewer #3: yes

**Summary and General Comments**

Reviewer #1: The manuscript submitted by M'Bondoukwé et al. is certainly very interesting because the trial they conducted compared the efficacy and safety of two albendazole regimens in individuals with fairly high Loa loa microfilarial densities. It deserves to be published in PLoS Neglected Tropical Diseases but a number of points should be addressed.

Reviewer #2: In their manuscript, the authors present the results of a randomized trial of 30 days of albendazole in hypermicrofilaremic loiasis. The question being addressed (i.e., what to do with individuals who are excluded form treatment using current approaches), is extremely important and this trial adds additional data to the body of published literature. The major strength of the paper is the inclusion of a relatively large cohort of hypermicrofilaremic patients. The manuscript is generally clear and well-written. Specific comments are provided above.

Reviewer #3: The authors report a clinical phase IIb trial assessing albendazole for the treatment of hypermicrofilaraemic loiasis in an highly endemic region in Gabon. The treatment of hypermicrofilaraemic loiasis is a much debated but unresolved issue and studies providing evidence for informed treatment decisions are therefore of high importance. The team is among the leading institutions in loiasis research worldwide and the study has been meticulously designed, implemented and analysed. The authors have to be congratulated for their work.

Main limitations of the study are the overall limited sample size and the short follow up period. These limitations are openly addressed by the authors. The conclusion of the paper is of high importance as it provides for the first time evidence for an optimsed treatment protocol for hypermicrofilaraemic loiasis in an endemic setting.

Please find my specific comments below:

"People living in Central Africa can carry thousands of these microscopic worms in their blood."

Suggest: People living in Central Africa can carry MILLIONS of these microscopic worms in their blood. (individuals may have thousands of mf per ml blood).

"help prepare patients for safer treatment with stronger medications and could improve control of this neglected tropical disease in hard-to-reach communities"

Suggest: safer treatment with curative medications

"SARs are common in nearly 10% of the exposed population"

This does not seem correct to me. Please reconsider

Inclusion/Exclusion criteria: better list them in Table than in the text

Sample size calculation: please specify the endpoint that was used

What were the most important non-eligibility reasons as there was a substantial proportion not include in the thrial?

Please provide SD for microfilaremia in Table 1

It is interesting to see that microfilaraemia increased slightly from day 21 in the ALB 400 group. This was also observed in the referenced trial (Manego et al 2023). It would be of interest if the authors could include this in the discussion.

"however, the ITT analysis suggested this occurred slightly earlier for the 400 mg group (Day 7)

and later for the 800 mg group (Day 11)."

This does not seem to correspond to the Figure. Please comment.

PLOS authors have the option to publish the peer review history of their article (what does this mean? ). If published, this will include your full peer review and any attached files.

**Do you want your identity to be public for this peer review?** For information about this choice, including consent withdrawal, please see our Privacy Policy .

Reviewer #1: No

Reviewer #2: No

Reviewer #3: No

**Figure resubmission:**

**Reproducibility:**



---

## [Decision Letter · Decision Letter 1]

23 Oct 2025

Assessment of the Efficacy and Safety of Two Albendazole Regimens for the treatment of hypermicrofilaraemic loiasis in adult populations in Woleu-Ntem Province, Gabon: a Phase IIb single-blind randomised controlled trial

Dear Dr. M'Bondoukwé,

Thank you for submitting your manuscript to PLOS Neglected Tropical Diseases. After careful consideration, we feel that it has merit but does not fully meet PLOS Neglected Tropical Diseases's publication criteria as it currently stands. Therefore, we invite you to submit a revised version of the manuscript that addresses the points raised during the review process.

Please submit your revised manuscript within 60 days Nov 22 2025 11:59PM. If you will need more time than this to complete your revisions, please reply to this message or contact the journal office at plosntds@plos.org. Please include the following items when submitting your revised manuscript:

We look forward to receiving your revised manuscript.

Kind regards,

Siddhartha Mahanty, M.B.B.S., M.P.H

Academic Editor

Jong-Yil Chai

Section Editor

Shaden Kamhawi

co-Editor-in-Chief

Paul Brindley

co-Editor-in-Chief

**Additional Editor Comments:**

The modifications have improved the manuscript considerably but, unfortunately, a number of the reviewers' valid concerns have not been addressed adequately or satisfactorily. In particular, issues raised by both reviewers 1 and 2 about the study rationale and assessment of microfilraemia are unresolved, for example the question of the accuracy of mcrofilaraemia, the primary outcome of the study and about the feasibility of using the proposed 30-day regimen of ABZ in the target populations. With the understanding that the manuscript reports on a clinical study and cannot be altered post hoc, the reviewers' concerns should be acknowledged and discussed in the context of limitations of the data and the study. For the manuscript to be accepted for publication please address or acknowledge with explanations all concerns raised by the reviewers particularly the issues with study design and parasitologic methodology discussed by reviewers 1 and 2.

**Journal Requirements:**

Please ensure that the CRediT author contributions listed for every co-author are completed accurately and in full.

At this stage, the following Authors/Authors require contributions: Noé Patrick M'Bondoukwé, Luccheri Ndong Akomezoghe, Bridy Chesly Moutombi Ditombi, Jacques Mari Ndong Ngomo, Hadry Roger Sibi Matotou, Ginette Sévérine Zang Ondo, Valentin Migueba, Coella Joyce Mihindou, Bedrich Pongui Ngondza, Christian Mayandza, Héléna Noéline Kono, Dimitri Ardin Moussavou Mabicka, Charleine Manomba, Reinne Moutongo, Luice Aurtin Joel James, Denise Patricia Mawili Mboumba, and Marielle Karine Bouyou Akotet. Please ensure that the full contributions of each author are acknowledged in the "Add/Edit/Remove Authors" section of our submission form.

**Reviewers' Comments:**

Reviewer's Responses to Questions

**Key Review Criteria Required for Acceptance?**

**Methods**

-Are the objectives of the study clearly articulated with a clear testable hypothesis stated?

-Is the study design appropriate to address the stated objectives?

-Is the population clearly described and appropriate for the hypothesis being tested?

-Is the sample size sufficient to ensure adequate power to address the hypothesis being tested?

-Were correct statistical analysis used to support conclusions?

-Are there concerns about ethical or regulatory requirements being met?

Reviewer #1: Non-minor comments on the Introduction and Methods sections:

- Line 55: do the authors really think that a 30-day course of albendazole can be applied "on a large scale"?

- Line 74: replace "beyond 8,000 mf/mL, the risk is 0.1% and around of 37.0% beyond 100,000 mf/mL" by "the risk is 0.1% for individuals with 8,000 mf/mL and around 37.0% in those with 100,000 mf/mL". In both cases, delete "beyond" : the values are for a GIVEN microfilarial density.

- Lines 87-88: same comment as above. Would long course of albendazole be administered at individual or community level?

- Figure 1.: the green dots, which were clearly visible on the first version of the figure, are no longer visible in the Haut-Ntem division after changing the colors

- Table 2: A clarification is needed for the second and third "exclusion criteria". Does this mean that the course of albendazole would be INTERRUPTED (participant’s withdrawal) if there were a "worsening of symptoms or the appearance of severity criteria related to L. loa or another disease" or an "increase in microfilaraemia of more than 50% compared to the initial value" ? If yes, which I suppose, the word "interruption" should be used somewhere

- Line 191: I still think that the word "control" is not appropriate because the results obtained in this group are not compared to those obtained with a similar group having benefitted from another intervention. Please delete "control" here (just write "the group with microfilaraemia below 8,000 mf/mL" and do the same throughout the text

- Line 390: what does "No reassessment of the microfilaremia was performed due to logistical constraints"? Do the authors mean "after D30"?

- Lines 394-413. The quantification of the microfilarial densities is probably much more accurate after leukoconcentration of 4 mL of blood than after examining 10 µL. Thus, I do not understand why both methods were used in parallel at each time point. In the Results section, which data were used ? Those obtained after leukoconcentration or those obtained after examination of 10 µL?

- Lines 440-442: the sentence is very long and confusing. Please split it in two sentences. Does "the reduction rates median" mean "the median of the reduction rates"

Reviewer #2: If I am understanding correctly, the microfilariae were quantified in 10 microliters of blood by counting motile microfilariae. Since many of the patients had very high levels of microfilaremia, this seems impossible to do with any accuracy. For example, the participant with 105,000 mf/mL would have 1050 mf/10 microliters of blood. Saponin lysis would be equally difficult to interpret as this patient would have >400,000 microfilariae on the slide. No information is provided re: how accuracy of the counts was ensured. Since the primary outcome of the study is mf count, this issue is essential to address. At the least, data should be provided on the accuracy and reproducibility of the counting method.

The reason for the cetirizine is still not addressed and is not standard procedure when administering albendazole. Although it is unlikely that this altered the metabolism of albendazole or the frequency of AEs (which typically occur once the mf begin to decrease), this should be clarified. The subheadings in the description of study drug administration could be reduced to a single sentence, assuming that administration procedures and use of cetirizine were comparable in all of the groups.

Reviewer #3: yes

**Results**

-Does the analysis presented match the analysis plan?

-Are the results clearly and completely presented?

-Are the figures (Tables, Images) of sufficient quality for clarity?

Reviewer #1: Comments on the Results and Discussion sections:

- Lines 521-523 : the sentence " Regarding the group of participants having received ALB 400 mg, there was equally a decrease in ITT (- 3825) mf/mL at D30) and PP approaches (16,950 (12,800 - 24975) mf/mL at D0 to 1,950 (675 – 7,325) mf/mL at D30)" needs to be revised

- Line 556, replace "superior" by "inferior" or "lower"

- Line 820, why do the authors mention Day 21 here ? (D21 was not a time point in their trial)

Reviewer #2: The values in Table 2 are very small and likely not normally distributed. They should be reported as median or geometric mean and “eosinophilia rate” should be defined.

Reviewer #3: yes

**Conclusions**

-Are the conclusions supported by the data presented?

-Are the limitations of analysis clearly described?

-Do the authors discuss how these data can be helpful to advance our understanding of the topic under study?

-Is public health relevance addressed?

Reviewer #1: Yes

Reviewer #2: The explanation of the more rapid reduction in mf in the present study compared to the prior study is inadequate. Other studies with albendazole in loiasis, including a large placebo controlled trial in Cameroon (PMID: 26967331) have not shown a rapid decrease in microfilarial levels with 800 mg dosing. It is hard to imagine how study design or monitoring could have led to the observed differences. A more likely explanation would be differences in the patient population (genetic differences in the CYP or rate of alcohol/tobacco use which induce these enzymes) and/or parasite (susceptibility to albendazole) between the study sites. Conversely, although I appreciate that the authors were responding to a reviewer comment, albendazole does not act directly on microfilariae and even in the 800 mg group did not lead to rapid clearing as is seen with DEC or ivermectin. Consequently, the potential risks of the higher dose now seem overstated.

Reviewer #3: yes

**Editorial and Data Presentation Modifications?**

Reviewer #1: Minor comments

- Everywhere in the text and the tables: please harmonize the thousands separators (with commas, eg 2,258, not 2258)

- Line 34, replace "microfilariaemia" by "microfilaraemia"

- Line 39, replace "microfilaremia" by "microfilaraemia" (and harmonize throughout the text

- Line 40 : replace "better susceptibility to" by "higher risk of"

- Line 79, delete "are not available" (these words are already at the end of the sentence)

- Line 84, replace "lead" by "contribute"

- Line 124: what does the "interruption of the Mectizan programme" mean?

- Line 311, replace "they received" by "the participants received"

- Line 323, delete "with the procedure supervised by the village community worker" (already said above)

- Line 504 : what does "1050-5425" correspond to? Minimum and maximum? Please clarify

- Line 507, add "hypermicrofilaraemic" before "participants" ?

- Line 512, add "in the geometric means" after "reductions" ?

- Line 513, what does "(-15.1-58.7)" correspond to ? Minimum and maximum ?

- Title of Fig. 4, the word "protocol" is missing

- Line 759, differences in the susceptibility of different "strains" of Loa loa could also be considered.

- Line 782, delete "While" ?

- Lines 811-813, merge the two sentences

Reviewer #2: “Unvisited” is not a standard term and is confusing (suggest that the team did not try to see them). Would use “Missed visit” instead which is the standard monitoring term and indicates only that the visit did not occur.

Figure 2 legend “correspond” should be “corresponds” , “administration” should be changed to “regimens” and the words “carried out” should be removed. There are also many English errors in the figure itself (and in the text added to the manuscript in general) that should be corrected.

I am completely confused by the following statement added in the section on patient followup: “No reassessment of the microfilaremia was performed due to logistical constraints.” This seems to directly contradict the preceding and following sentences which talks about sampling at D2, 7, 14 and 30.

Reviewer #3: (No Response)

**Summary and General Comments**

Reviewer #1: The authors have well addressed the comments made by the reviewers.

The results of this trial are most interesting and should be published as soon as possible.

However, the authors should still make some changes before the manuscript can be definitively be accepted.

Reviewer #2: In their revised manuscript, the authors present the results of a randomized trial of 30 days of

albendazole in hypermicrofilaremic loiasis. The question being addressed (i.e., what to do with

individuals who are excluded form treatment using current approaches), is extremely important and

this trial adds additional data to the body of published literature. The major strength of the paper is the

inclusion of a relatively large cohort of hypermicrofilaremic patients. The manuscript is generally clear

and well-written. Specific comments are provided above.

Reviewer #3: The authors have adequately addressed my comments. The table depicting inclusion and exclusion criteria seems somewhat complicated (having the categories inclusion, non-inclusion, exclusion) - I suggest to simplify this table into "inclusion" and "exclusion" criteria only.

PLOS authors have the option to publish the peer review history of their article (what does this mean? ). If published, this will include your full peer review and any attached files.

**Do you want your identity to be public for this peer review?** For information about this choice, including consent withdrawal, please see our Privacy Policy .

Reviewer #1: No

Reviewer #2: No

Reviewer #3: No

**Figure resubmission:**
---

## [Decision Letter · Decision Letter 2]

23 Dec 2025

Assessment of the Efficacy and Safety of Two Albendazole Regimens for the Treatment of Hypermicrofilaraemic Loiasis in Adults in Woleu-Ntem Province, Gabon: a Phase IIb Single-Blind Randomised Controlled Trial

Dear Dr. M'Bondoukwé,

Thank you for submitting your manuscript to PLOS Neglected Tropical Diseases. After careful consideration, we feel that it has merit but does not fully meet PLOS Neglected Tropical Diseases's publication criteria as it currently stands. Therefore, we invite you to submit a revised version of the manuscript that addresses the points raised during the review process.

* A letter that responds to each point raised by the editor and reviewer(s). You should upload this letter as a separate file labeled 'Response to Reviewers '. This file does not need to include responses to any formatting updates and technical items listed in the 'Journal Requirements' section below.

* A marked-up copy of your manuscript that highlights changes made to the original version. You should upload this as a separate file labeled 'Revised Manuscript with Track Changes '.

* An unmarked version of your revised paper without tracked changes. You should upload this as a separate file labeled 'Manuscript '.

We look forward to receiving your revised manuscript.

Kind regards,

Siddhartha Mahanty, M.B.B.S., M.P.H

Academic Editor

Jong-Yil Chai

Section Editor

Shaden Kamhawi

co-Editor-in-Chief

Paul Brindley

co-Editor-in-Chief

**Additional Editor Comments:**

For acceptance of the manuscript for publication please respond to the minor issues raised by the reviewers, particularly those mentioned by reviewer 2 relating to study withdrawal, the accuracy of microfilaraemia and adverse effects of albendazole.

**Journal Requirements:**

At this stage, the following Authors/Authors require contributions: Noé Patrick M'Bondoukwé, Luccheri Ndong Akomezoghe, Bridy Chesly Moutombi Ditombi, Jacques Mari Ndong Ngomo, Hadry Roger Sibi Matotou, Ginette Sévérine Zang Ondo, Valentin Migueba, Coella Joyce Mihindou, Bedrich Pongui Ngondza, Christian Mayandza, Héléna Noéline Kono, Dimitri Ardin Moussavou Mabicka, Charleine Manomba, Reinne Moutongo, Luice Aurtin Joel James, Denise Patricia Mawili Mboumba, and Marielle Karine Bouyou Akotet. Please ensure that the full contributions of each author are acknowledged in the "Add/Edit/Remove Authors" section of our submission form.

**Reviewers' comments:**

Reviewer's Responses to Questions

**Key Review Criteria Required for Acceptance?**

**Methods**

-Are the objectives of the study clearly articulated with a clear testable hypothesis stated?

-Is the study design appropriate to address the stated objectives?

-Is the population clearly described and appropriate for the hypothesis being tested?

-Is the sample size sufficient to ensure adequate power to address the hypothesis being tested?

-Were correct statistical analysis used to support conclusions?

-Are there concerns about ethical or regulatory requirements being met?

Reviewer #1: OK

Reviewer #2: The change to Table 1 has made it more confusing – withdrawal implies interruption of treatment but not the converse. Were there events that could lead to interruption of treatment but not study withdrawal? If not, the title is OK and could be shortened to “withdrawal or treatment interruption”. If there are differences, these should be indicated or listed in two separate columns (study withdrawal, treatment interruption).

The quantification of parasitemia is much clearer now. However, dilution amplifies errors (a single microfilaria difference can lead to a thousand-fold difference in count at a dilution of 1:1000). I realize that this may be impossible at this stage but it would be helpful to know the accuracy of the method used (if two people look at the same blood sample, how much variation is there is the count).

**Results**

-Does the analysis presented match the analysis plan?

-Are the results clearly and completely presented?

-Are the figures (Tables, Images) of sufficient quality for clarity?

Reviewer #1: OK

Reviewer #2: Comments have been adequately addressed in revision.

**Conclusions**

-Are the conclusions supported by the data presented?

-Are the limitations of analysis clearly described?

-Do the authors discuss how these data can be helpful to advance our understanding of the topic under study?

-Is public health relevance addressed?

Reviewer #1: OK

Reviewer #2: The last statement in the abstract about the potential for serious adverse events should be removed. Albendazole has no direct microfilaricidal effect (the reason for the adverse effects following ivermectin and DEC). This is supported by the finding in prior studies of albendazole and by the data in this paper (see figure 6 and 7 which effectively show little or no difference in post-treatment symptoms between the groups that did and did not receive albendazole). Although the reduction in microfilaremia is slightly more rapid in this study that in the prior studies, it is still much slower and less complete than what is seen with microfilaricidal drugs, consistent with the mechanism of action on adult worms. The rare cases of encephalopathy in the literature may or may not be due to albendazole (encephalopathy has been described in loiasis in the absence of treatment and the conclusions are based solely on a temporal association). These are given too much weight in the discussion. Prolonged treatment with albendazole is, however, associated with other adverse events such as elevated transaminases, cytopenias, and Stevens-Johnson syndrome that are rare but significant (as mentioned in the discussion).

**Editorial and Data Presentation Modifications?**

Reviewer #1: OK

Reviewer #2: There are numerous remaining grammar, spelling, typographical errors. Examples:

In Figure 1 the use of red lettering on a green background will prevent the 10% of males who are red-green colorblind form being able to see the letters.

Line 255 – “all” and “the” are reversed and there is an extra “-or” at the end of the sentence.

Line 261 - The statement “no reassessment of microfilaremia was performed” is still confusing. Do the authors mean “no reassessment of microfilaremia was performed after D30” – if so, that is how it should be stated. As written it conflicts with the prior sentence describing the reassessment of microfilaremia.

Line 267 – microfilaraemia is misspelled (micrrofilraemia)

Line 273 – the verb (“performed”) is missing.

Line 290 – there is a period missing.

“Eosinophilia rate” is an incorrect term. This would mean the frequency of individuals with an elevated blood eosinophil count. That is not what is described in the methods. The correct heading would be “quantification of eosinophilia”.

**Summary and General Comments**

Reviewer #1: OK

Reviewer #2: The authors have responded to the majority of the reviewers' comments. Remaining issues are outlined above.

PLOS authors have the option to publish the peer review history of their article (what does this mean? ). If published, this will include your full peer review and any attached files.

**Do you want your identity to be public for this peer review?** For information about this choice, including consent withdrawal, please see our Privacy Policy .

Reviewer #1: No

Reviewer #2: No

**Figure resubmission:**
---

## [Editor Report · Decision Letter 3]

30 Dec 2025

Dear Dr M'Bondoukwé,

We are pleased to inform you that your manuscript 'Assessment of the Efficacy and Safety of Two Albendazole Regimens for the Treatment of Hypermicrofilaraemic Loiasis in Adults in Woleu-Ntem Province, Gabon: a Phase IIb Single-Blind Randomised Controlled Trial' has been provisionally accepted for publication in PLOS Neglected Tropical Diseases.

Best regards,

Jong-Yil Chai

Section Editor

Jong-Yil Chai

Section Editor

Shaden Kamhawi

co-Editor-in-Chief

Paul Brindley

co-Editor-in-Chief

The 2nd-time revised manuscript is now acceptable by the journal. Thanks for your kind cooperation.

---

## [Editor Report · Acceptance letter]

Dear Dr M'Bondoukwé,

We are delighted to inform you that your manuscript, "

Assessment of the Efficacy and Safety of Two Albendazole Regimens for the Treatment of Hypermicrofilaraemic Loiasis in Adults in Woleu-Ntem Province, Gabon: a Phase IIb Single-Blind Randomised Controlled Trial," has been formally accepted for publication in PLOS Neglected Tropical Diseases.

Best regards,

Shaden Kamhawi

co-Editor-in-Chief

Paul Brindley

co-Editor-in-Chief
